# LEARNING STRUCTURAL EDITS VIA INCREMENTAL TREE TRANSFORMATIONS

**Ziyu Yao**[*]
The Ohio State University
yao.470@osu.edu

**Frank F. Xu, Pengcheng Yin**
Carnegie Mellon University
{fangzhex,pcyin}@cs.cmu.edu

**Huan Sun**
The Ohio State University
sun.397@osu.edu

**Graham Neubig**
Carnegie Mellon University
gneubig@cs.cmu.edu

## ABSTRACT

While most neural generative models generate outputs in a single pass, the human creative process is usually one of iterative building and refinement. Recent work has proposed models of editing processes, but these mostly focus on editing sequential data and/or only model a single editing pass. In this paper, we present a generic model for incremental editing of structured data (*i.e.* "structural edits"). Particularly, we focus on tree-structured data, taking abstract syntax trees of computer programs as our canonical example. Our editor learns to iteratively generate tree edits (*e.g.* deleting or adding a subtree) and applies them to the partially edited data, thereby the entire editing process can be formulated as consecutive, incremental tree transformations. To show the unique benefits of modeling tree edits directly, we further propose a novel edit encoder for learning to represent edits, as well as an imitation learning method that allows the editor to be more robust. We evaluate our proposed editor on two source code edit datasets, where results show that, with the proposed edit encoder, our editor significantly improves accuracy over previous approaches that generate the edited program directly in one pass. Finally, we demonstrate that training our editor to imitate experts and correct its mistakes dynamically can further improve its performance.

## 1 INTRODUCTION

Iteratively revising existing data for a certain purpose is ubiquitous. For example, researchers repetitively polish their manuscript until the writing becomes satisfactory; computer programmers keep editing existing code snippets and fixing bugs until desired programs are produced. Can we properly model such iterative editing processes with neural generative models?

To answer this question, previous works have examined models for editing sequential data such as natural language sentences. Some example use cases include refining results from a first-pass text generation system (Simard et al., 2007; Xia et al., 2017), editing retrieved text into desired outputs (Gu et al., 2018; Guu et al., 2018), or revising a sequence of source code tokens (Yin et al., 2019; Chen et al., 2019; Yasunaga & Liang, 2020). These examples make a single editing pass by directly generating the edited sequence. In contrast, there are also works on modeling the *incremental edits* of sequential data, which predict sequential edit operations (*e.g.* keeping, deleting or adding a token) either in a single pass (Shin et al., 2018; Vu & Haffari, 2018; Malmi et al., 2019; Dong et al., 2019; Stahlberg & Kumar, 2020; Iso et al., 2020) or iteratively (Zhao et al., 2019; Stern et al., 2019; Gu et al., 2019a;b), or modify a sequence in a non-autoregressive way (Lee et al., 2018).

However, much interesting data in the world has strong underlying structure such as trees. For example, a syntactic parse can be naturally represented as a tree to indicate the compositional relations among constituents (*e.g.* phrases, clauses) in a sentence. A computer program inherently is also a tree defined by the programming language's syntax. In the case that this underlying structure exists, many edits can be expressed much more naturally and concisely as transformations over the underlying trees than conversions of the tokens themselves. For example, removing a statement from a

---

[*]Work done while interning at CMU.

computer program can be easily accomplished by deleting the corresponding tree branch as opposed to deleting tokens one by one. Despite this fact, work on editing tree-structured data has been much more sparse. In addition, it has focused almost entirely on single-pass modification of structured outputs as exemplified by Yin et al. (2019); Chakraborty et al. (2020) for computer program editing.

In this work, we are interested in a generic model for *incremental* editing of structured data (*"structural edits"*). Particularly, we focus on *tree-structured* data, taking abstract syntax trees of computer programs as our canonical example. We propose a neural editor that runs iteratively. At each step, the editor *generates* and *applies* a tree edit (*e.g.* deleting or adding a subtree) to the partially edited tree, which deterministically transforms the tree into its modified counterpart. Therefore, the entire tree editing process can be formulated as consecutive, incremental tree transformations (Fig. 1).

While recent works (Tarlow et al., 2019; Dinella et al., 2020; Brody et al., 2020) have also examined models that make changes to trees, our work is distinct from them in that: *First*, compared with Dinella et al. (2020), we studied a different problem of editing tree-structured data particularly triggered by an *edit specification* (which implies a certain edit intent such as a code refactoring rule). *Second*, we model structural edits via *incremental tree transformations*, while Tarlow et al. (2019) and Brody et al. (2020) predict a complete edit sequence based on the fixed input tree, without applying the edits or performing any tree transformations incrementally. Although Dinella et al. (2020) have explored a similar idea, our proposed tree editor is more general owing to the adoption of the Abstract Syntax Description Language (ASDL; Wang et al. (1997)). This offers our editor two properties: being *language-agnostic* and ensuring *grammar validity*. In contrast, Dinella et al. (2020) include JavaScript-specific design and employ only ad-hoc grammar checking. *Finally*, our tree editor supports a comprehensive set of operations such as adding or deleting a tree node and copying a subtree, which can fulfill a broad range of tree editing requirements. These operations are not fully allowed by previous work, *e.g.*, Brody et al. (2020) cannot add (or generate) a new tree node from scratch; Tarlow et al. (2019) and Dinella et al. (2020) do not support subtree copying.

We further propose two modeling and training improvements, specifically enabled by and tailored to our incremental editing formalism. First, we propose a new *edit encoder* for learning to represent the edits to be performed. Unlike existing edit encoders, which compress tree differences at the token level (Yin et al., 2019; Hoang et al., 2020; Panthaplackel et al., 2020b) or jointly encode the initial and the target tree pairs in their surface forms (Yin et al., 2019), our proposed edit encoder learns the representation by encoding the sequence of gold tree edit actions. Second, we propose a novel *imitation learning* (Ross et al., 2011) method to train our editor to correct its mistakes dynamically, given that it can modify any part of a tree at any time.

We evaluate our proposed tree editor on two source code edit datasets (Yin et al., 2019). Our experimental results show that, compared with previous approaches that generate the edited program in one pass, our editor can better capture the underlying semantics of the intended edits, which allows it to outperform existing approaches by more than 7% accuracy in a one-shot evaluation setting. With the proposed edit encoder, our editor significantly improves accuracy over existing state-of-the-art methods on both datasets. We also demonstrate that our editor can become more robust by learning to imitate expert demonstrations dynamically. Our source code is available at `https://github.com/neulab/incremental_tree_edit`.

## 2    PROBLEM FORMULATION

As stated above, our goal is to create a general-purpose editor for tree-structured data. Specifically, we are interested in editing tree structures defined following an underlying grammar that, for every parent node type, delineates the allowable choices of child nodes. Such *syntactic* tree structures, like syntax trees of sentences or computer programs, are ubiquitous in fields like natural language processing and software engineering. In this paper we formulate editing such tree structures as revising an input tree $C_-$ into an output tree $C_+$ according to an edit specification $\Delta$. As a concrete example, we use editing abstract syntax trees (ASTs) of C# programs, as illustrated in Fig. 1. This figure shows transforming the AST of "`x=list.ElementAt(i+1)`" ($C_-$) to the AST of "`x=list[i+1]`" ($C_+$). In this case, the edit specification $\Delta$ could be interpreted as a refactoring rule that uses the bracket operator $[\,\cdot\,]$ for accessing elements in a list.[1] In practice, the edit specification is learned

---

[1] The corresponding Roslyn analyzer in C# can be found at `https://github.com/JosefPihrt/Roslynator/blob/master/docs/analyzers/RCS1246.md`.

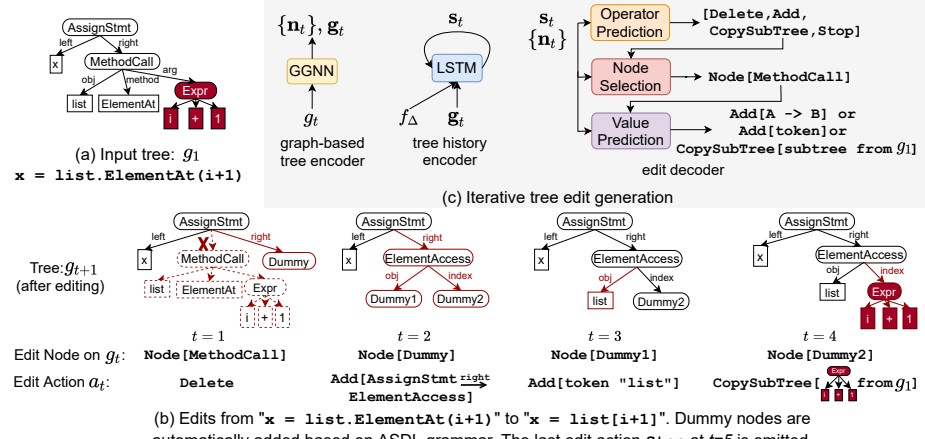

(a) Input tree: $g_1$
`x = list.ElementAt(i+1)`

(c) Iterative tree edit generation

(b) Edits from "`x = list.ElementAt(i+1)`" to "`x = list[i+1]`". Dummy nodes are automatically added based on ASDL grammar. The last edit action `Stop` at $t=5$ is omitted.

Figure 1: Our proposed neural editor for editing tree-structured data.

by an *edit encoder* $f_\Delta$ from a pair of input-output examples $\langle C'_-, C'_+ \rangle$, and encoded as a real-valued edit representation, *i.e.* $f_\Delta(C'_-, C'_+) \in \mathbb{R}^n$. The learned edit representation could then be used to modify $C_-$ in a similar way as editing $C'_-$. Onwards we use $f_\Delta$ as a simplified notation for edit representations.

Revising a tree into another typically involves a sequence of incremental edits. For instance, to modify the input tree in the aforementioned example, one may first delete the subtree rooted at the node `MethodCall`, which corresponds to the code fragment "`list.ElementAt(i+1)`", and then replace it with an `ElementAccess` node denoting the bracket operator, etc. We formulate this editing process as a sequential decision making process $(\langle g_1, a_1 \rangle, \ldots, \langle g_T, a_T \rangle)$, where for each tree $g_t$ at time step $t$, the editor executes a tree edit action $a_t$, deterministically transforming it into $g_{t+1}$. In particular, $g_1$ is the initial input tree $C_-$.[2] The process stops at $g_T$ when the editor predicts a special `Stop` action as $a_T$. Denoting $g_{1:t} = (g_1, ..., g_t)$ as the tree history and $a_{1:t} = (a_1, ..., a_t)$ the edit history until step $t$, then the editing can be framed as the following autoregressive process:

$$p(a_{1:T}|f_\Delta, g_1) = p(a_1|f_\Delta, g_1)p(a_2|f_\Delta, g_{1:2}) \cdots p(a_T|f_\Delta, g_{1:T}) = \prod_{t=1}^{T} p(a_t|f_\Delta, g_{1:t}). \quad (1)$$

## 3 MODEL

We will introduce our neural editor for modeling $p(a_t|\cdot)$ in § 3.1, followed by the edit representation model $f_\Delta$ in § 3.2.

### 3.1 NEURAL TREE EDITOR

Fig. 1(c) illustrates our editor architecture. At each time step, the editor first encodes the current tree $g_t$ and the tree history $g_{1:t}$. It then employs a modular decoder to predict a tree edit action $a_t$. Next, we will first introduce our tree edit actions and then elaborate the model details.

### 3.1.1 TREE EDIT ACTIONS

Our editor uses a sequence of editing actions to incrementally modify a tree-structured input. At each time step, the decoder takes an action $a_t$ to update a partially-edited tree $g_t$. Specifically, an action $a_t$ consists of an operator (*e.g.* an operator that removes a subtree from $g_t$) with its optional arguments (*e.g.* the target subtree to delete). Importantly, the space of actions is limited to maintain consistency with the underlying syntax of the language. While a number of syntactic formalisms such as context free grammar (Chomsky, 1956) or tree substitution grammar (Cohn et al., 2010) exist, in this work we choose the ASDL formalism due to its ability to flexibly handle optional and sequential fields (interested readers may reference Wang et al. (1997) and Yin & Neubig (2018) for details). Under this framework, we define four types of operators.

---

[2]Notably, the special case of empty initial trees corresponds to code generation from scratch. Thus our formulation applies to *both* tasks of editing existing trees and generating new ones.

`Delete` operators take a tree node $n_t$ as argument and remove $n_t$ and its descendants from $g_t$ (*e.g.* $t = 1$ in Fig. 1(b)). Note that removing arbitrary (child) nodes from $g_t$ might produce syntactically invalid trees, since under the grammar, a parent node type would always have a fixed set of edge types. For instance, if the node `MethodCall` and its incoming edge `right` were to be removed at $t = 1$, the resulting AST would be syntactically invalid under C#'s grammar, as the node `AssignStmt` denoting a variable assignment statement is missing a child node representing its right operand. To maintain syntactic correctness (no missing child nodes for any parent nodes), we therefore replace the to-be-deleted node with a pseudo `Dummy` node as a placeholder.

Next, we define an `Add` operator to add new nodes to $g_t$. The operator first locates a target position by selecting an existing tree node. We consider two cases based on the edge type (or "field") of the target position: for *single* or *optional* fields that allow at most one child (*e.g.* field `right` in Fig. 1(b) at $t = 1$), the selected tree node has to be their dummy child node (*e.g.* node `Dummy` at $t = 1$) and the `Add` operator will then *replace* the dummy node with the new tree node; for *sequential* fields that accept more than one child (*e.g.* the field of a "statement block" that allows an arbitrary number of statements), the selected tree node can be any child node of the field (including a rightmost dummy node we append to every sequential field) and the `Add` operator will then *insert* the new node before the selected node. We elaborate this mechanism in § A.1. For our editor, adding a *non-terminal* node (*e.g.* node `ElementAccess` in Fig. 1(b) at $t = 2$) is equivalent to selecting a production rule to derive its field (*e.g.* `AssignStmt` $\xrightarrow{\text{right}}$ `ElementAccess`). As with `Delete` actions, to ensure there is no missing child node, we instantiate the set of child nodes with dummy nodes for the newly added node based on the underlying grammar, which leads to nodes `Dummy1` and `Dummy2` at $t = 2$. `Add` can also be used to populate empty *terminal* nodes with actual values (*e.g.* string token "`list`" at $t = 3$). This is the same as picking a token from the token vocabulary.

Additionally, observing that in many cases, revising a tree can be easily done by copying a subtree from the initial input $g_1$ (*e.g.* subtree `Expr` $\mapsto$ `i + 1` in Fig. 1(a)) to a new position in the updated tree $g_t$ (*e.g.* the right child position of node `ElementAccess` in Fig. 1(b) at $t = 4$), we introduce a high-level operator `CopySubTree`. This operator locates a target position similarly as the `Add` operator and then copies a *complete* subtree from $g_1$ to the target position in a single step.

Finally, a `Stop` action is used to terminate the iterative tree editing procedure, after which the remaining dummy nodes will be cleared. We note that our framework has decoupled the language grammar specifications (handled by ASDL) from the model architecture (corresponding to our language-agnostic model implementation), and thus can be applied to various languages flexibly.

### 3.1.2 Tree and Tree History Encoder

Similarly to existing works in learning tree representations (Allamanis et al., 2018; Brockschmidt et al., 2018; Yin et al., 2019; Hellendoorn et al., 2020), we adopt a graph-based encoder to learn representations of each tree $g_t$. Specifically, we follow Allamanis et al. (2018), and extend $g_t$ into a graph by adding bidirectional edges between parent and child nodes, as well as adjacent sibling nodes. We use a gated graph neural network (GGNN, Li et al. (2015)) to compute a vector representation $\boldsymbol{n}_t$ for each node $n_t$ on tree $g_t$, and mean-pool $\{\boldsymbol{n}_t\}$ to represent $g_t$, denoted $\boldsymbol{g}_t$.

An LSTM encoder is used to track the tree history $g_{1:t}$, *i.e.* $\boldsymbol{s}_t = \text{LSTM}([\boldsymbol{g}_t; f_\Delta], \boldsymbol{s}_{t-1})$, where $[\cdot; \cdot]$ denotes vector concatenation. We will introduce how to learn the edit representation $f_\Delta$ in § 3.2. The updated state $\boldsymbol{s}_t$ is then used to predict edit actions, as elaborated next.

### 3.1.3 Tree Edit Decoder

Our edit decoder predicts an action $a_t$ using three components: an operator predictor, a node selector, and a value predictor. At each time step $t$, the decoder's operator predictor first decides which operator $op_t \in \{\texttt{Delete}, \texttt{Add}, \texttt{CopySubTree}, \texttt{Stop}\}$ to apply. Next, for operators other than `Stop`, the node selector predicts a node $n_t$ from the tree to locate the target position for applying $op_t$. Finally, if $op_t \in \{\texttt{Add}, \texttt{CopySubTree}\}$, the value predictor further determines additional arguments of those operators (denoted as $val_t$, *e.g.* the to-be-added node for `Add`). This is summarized as:

$$p(a_t|\boldsymbol{s}_t) = p(op_t|\boldsymbol{s}_t)p(n_t|\boldsymbol{s}_t, op_t)p(val_t|\boldsymbol{s}_t, op_t, n_t). \tag{2}$$

**Operator Prediction:** The operator prediction is a 4-class classification problem. We calculate the probability of taking operator $op_t$ as $p(op_t|\boldsymbol{s}_t) = \text{softmax}(\boldsymbol{W}_{op}\boldsymbol{s}_t + \boldsymbol{b}_{op})$.

**Node Selection:** Given a tree $g_t$, there could exist an arbitrary number of tree nodes. Therefore, we design the node selection module similar to a pointer network (Vinyals et al., 2015). To this end, we learn a hidden state $\boldsymbol{h}_{node,t} = \tanh(\boldsymbol{W}_{node}[\boldsymbol{s}_t; \text{emb}(op_t)] + \boldsymbol{b}_{node})$ as the "pointer", where $\text{emb}(op_t)$ embeds the previously selected operator $op_t$. In our model, "emb($\cdot$)" denotes learnable embeddings. We then calculate the inner product of $\boldsymbol{h}_{node,t}$ and each node representation $\boldsymbol{n}_t$ for node selection.

**Value Prediction:** The value predictor predicts an argument $val_t$ for `Add` and `CopySubTree` actions. For `Add` actions, $val_t$ denotes the new tree node (corresponding to a production rule or a terminal token) to be added to $g_t$. For `CopySubTree` actions, $val_t$ is the subtree from $g_1$ to be copied to $g_t$. In both cases, we only consider the candidate set $\{val_t\}$ allowable under the grammar constraints. Similarly to the node predictor, the distribution $p(val_t|\cdot)$ is also given by a pointer network, with its hidden state defined as $\boldsymbol{h}_{val,t} = \tanh(\boldsymbol{W}_{val}[\boldsymbol{s}_t; \boldsymbol{n}_t; \text{emb}(p_{n_t} \mapsto n_t)] + \boldsymbol{b}_{val})$, where $\text{emb}(p_{n_t} \mapsto n_t)$ is the embedding of the edge type between the parent node $p_{n_t}$ and the child $n_t$ (*e.g.* field `right` for `AssignStmt` $\xrightarrow{\text{right}}$ `ElementAccess`). Depending on the type of $val_t$, its representation could be either a learned embedding of the production rule, a word embedding, or a subtree encoding given by the representation of its root node. We refer readers to § A.2 for details.

## 3.2 Tree Edit Encoding

Given an edit pair $\langle C_-, C_+ \rangle$, we aim to learn a real-valued vector $f_\Delta(C_-, C_+)$ to represent the intent behind the edits. This is a crucial task and has been investigated in several previous works. For example, Yin et al. (2019), Panthaplackel et al. (2020b), and Hoang et al. (2020) considered edits at the *token level* and used either a bag-of-edits encoder or a sequence encoder to encode the differences between $C_-$ and $C_+$. As a result, these edit encoders have abandoned the syntactic structure of tree edits. Yin et al. (2019) further proposed a graph edit encoder, which connects the input and output trees with labeled edges such as "Removed" and "Added", and then encodes the connected trees via a graph neural network. Although structural tree differences have been expressed in this edit encoder, the differences are modeled rather *implicitly* as the edit encoder has simply treated them as additional graph features.

In this section, we present a novel edit encoder which instead shifts the modeling focus completely to the *targeted edit actions* themselves. Specifically, it learns an edit representation by directly encoding the sequence of structural edit actions $(a_1, a_2, ..., a_T)$ that transforms $C_-$ to $C_+$. The encoder first computes a representation $\boldsymbol{a}_t$ for each action $a_t$, depending on the type of its operator:

$$\boldsymbol{a}_{\texttt{Stop}} = \boldsymbol{W}_{\texttt{Stop}}\text{emb}(\texttt{Stop}) + \boldsymbol{b}_{\texttt{Stop}},$$
$$\boldsymbol{a}_{\texttt{Delete}} = \boldsymbol{W}_{\texttt{Delete}}[\text{emb}(\texttt{Delete}); \boldsymbol{n}_t; \text{emb}(p_{n_t} \mapsto n_t)] + \boldsymbol{b}_{\texttt{Delete}},$$
$$\boldsymbol{a}_{\texttt{Add}} = \boldsymbol{W}_{\texttt{Add}}[\text{emb}(\texttt{Add}); \boldsymbol{n}_t; \text{emb}(p_{n_t} \mapsto n_t); \text{emb}(val_t)] + \boldsymbol{b}_{\texttt{Add}},$$
$$\boldsymbol{a}_{\texttt{CopySubTree}} = \boldsymbol{W}_{\texttt{CopySubTree}}[\text{emb}(\texttt{CopySubTree}); \boldsymbol{n}_t; \text{emb}(p_{n_t} \mapsto n_t); \text{emb}(subtree_t)] + \boldsymbol{b}_{\texttt{CopySubTree}}.$$

The proposed edit encoder then feeds the sequence of action representations $\{\boldsymbol{a}_t\}_{t=1}^T$ into a bidirectional LSTM, whose last hidden state is used as the edit representation $f_\Delta(C_-, C_+)$.

## 3.3 Training and Inference

We jointly train the proposed editor and the edit encoder in an autoencoding style, following Yin et al. (2019). Specifically, given an edit pair $\langle C_-, C_+ \rangle$ in the training set, we assume a gold-standard edit action sequence $a_{1:T}^*$ which edits $C_-$ to $C_+$ (*e.g.* the edit action sequence in Fig. 1). We seek to maximize the probability of $p(a_{1:T}^*|f_\Delta(C_-, C_+), C_-)$ in training.[3] By decomposing the probability according to Eq. (2), this is equivalent to jointly maximizing the probability of each edit decoder module making the gold decision at each time step. In practice, we use dynamic programming (pseudo code can be found in § C.1) to calculate the shortest tree edit sequence as $a_{1:T}^*$,[4] and compute a cross entropy loss for each edit decoder module.

---

[3] $f_\Delta(C_-, C_+)$ is one real-valued vector and thus does not *directly* expose $C_+$. We set it to a low dimension following Yin et al. (2019), which bottlenecks the vector's ability to memorize the entire output.

[4] We assume a left-to-right, top-down order when comparing the input/output tree. Future work can also consider other orders to improve the editing quality (Gu et al., 2019a; Welleck et al., 2019; Góis et al., 2020).

At inference time, given an input tree $C_-$ and an edit representation $f_\Delta$ (calculated either from $\langle C_-, C_+ \rangle$ or another edit pair $\langle C'_-, C'_+ \rangle$), we generate one tree edit at each time step $t$ by greedily deciding the operator, the node and the value. The generated edit is then applied to the tree so it transits to $g_{t+1}$ deterministically. We then update the tree history representation $s_{t+1}$ for generating the next tree edit. The inference process ends when a `Stop` operator is chosen.

## 4 ROBUST STRUCTURAL EDITING VIA IMITATION LEARNING

A unique advantage that distinguishes our editor from existing ones is its potential to fix wrong edits and iteratively refine its *own* output. This is achievable because our editor can revise any part of a tree at any time. We investigate this hypothesis by training the proposed editor via imitation learning, where the editor learns to imitate gold edit actions ("expert demonstrations") under *states* it visits at the inference time. Here, we define a "state" $s_t$ to include the current tree history $g_{1:t}$ and the edit representation $f_\Delta$. Our learning algorithm follows DAGGER (Ross et al., 2011), where in each training iteration, for a given $\langle f_\Delta, C_-, C_+ \rangle$ tuple, we first run the editor to infer and apply a sequence of edits resulting in a "trajectory" of $(\langle s_1, a_1 \rangle, ..., \langle s_T, a_T \rangle)$. We then request a gold edit action $\pi^*(s_t)$ for each state $s_t$ visited by the editor. The collected state-gold edit action pairs are aggregated to retrain the editor for the next iteration. This sampling and demonstration collecting strategy (denoted as DAGGERSAMPLING) is shown in Algo. 1 (Appendix B). Note that, in practice, instead of sampling a trajectory solely from the learning editor $\pi_\theta$, the DAGGER algorithm samples from a mixture policy $\pi'$, with which the actual edit action $a_t$ at each step $t$ comes from either $\pi_\theta$ with a probability of $1 - \beta$ or the "expert" $\pi^*$ with a probability of $\beta$.

To simulate the "expert", we calculate "dynamic oracles" (Goldberg & Nivre, 2012) by comparing the current tree with the target output tree. For example, in Fig. 1, if our editor incorrectly takes "Add[AssignStmt $\mapsto$ Expr]" at $t = 2$, the dynamic oracle will produce "Delete[AssignStmt $\rightarrow$ Expr]" as the gold edit action at $t = 3$ to revoke the wrong edit. This thus provides a means for the editor to learn to correct mistakes that it will likely produce at inference time.

Preliminary results showed that the editor trained following DAGGERSAMPLING may fall into a loop of repetitively deleting and adding the same component. We hypothesize that teaching the editor to imitate experts under unstable states (*i.e.* amid its initial full pass of editing) could be detrimental. Therefore, we propose another sampling strategy, POSTREFINESAMPLING, which samples and collects state-action pairs from the *expert* as a *post refinement* step (Algo. 2 in Appendix B). Specifically, we first run our editor to finish its sequential editing, which gives the output tree $g_T$ (Line 2). If $g_T$ is different from target $C_+$, we run the expert policy $\pi^*$ to continue editing until it successfully reaches $C_+$, and return state-action pairs collected from the expert as training material for the editor (Line 3-5). When $g_T$ is correct, no further training data will be collected (Line 6-8).

## 5 EXPERIMENTS

### 5.1 EXPERIMENTAL SETUP

We test our methods on two source code edit datasets introduced by Yin et al. (2019), also largely following their experimental setting.

The GitHubEdits (GHE) dataset contains $\langle C_-, C_+ \rangle$ pairs and their surrounding context collected from the commit logs of 54 GitHub C# projects. The dataset is split into train/dev/test sets of 91,372 / 10,176 / 10,176 samples. We jointly learn an edit representation $f_\Delta(C_-, C_+)$ while training the editor to generate $C_+$ from $C_-$. In evaluation, we measure the accuracy of each editor based on whether they successfully edit $C_-$ to the exact gold $C_+$. Since the edit representation $f_\Delta$ is calculated from the targeted $\langle C_-, C_+ \rangle$ pair, we denote this setting as **GHE-gold**.

The second dataset, C#Fixers (Fixers), is relatively small, containing 2,878 $\langle C_-, C_+ \rangle$ pairs. Unlike GHE, edit pairs in Fixers are built using 16 C# "fixers" with known semantics (*e.g.* removing redundant parentheses as a way to perform refactoring). As standard, we use this dataset only for *testing* purposes (*i.e.* all methods are first *trained* on GHE-gold). We consider a **Fixers-gold** setting similar as GHE-gold to evaluate the accuracy of generating $C_+$ from $\langle f_\Delta(C_-, C_+), C_- \rangle$.

Since edits in Fixers have known semantics, we also use Fixers to test methods in a one-shot setting (denoted as **Fixers-one shot**): For each $\langle C_-, C_+ \rangle$ pair, we select another $\langle C'_-, C'_+ \rangle$ pair from the

same fixer category (which bears the same edit intent as $\langle C_-, C_+ \rangle$ but is applied to a different input) to infer the edit representation and evaluate the accuracy of generating $C_+$ from $\langle f_\Delta(C'_-, C'_+), C_- \rangle$. We follow Panthaplackel et al. (2020a) and pick the first 100 samples at most per fixer category (the "seeds"), compute an edit representation of each one, and apply it to edit the others. We then report an average accuracy over the seeds as the score for this fixer category. Because the sample size of each fixer category is highly imbalanced, we report both *macro average* (treating all categories equally) and *micro average* (dependent on the sample size) edit accuracies over the 16 fixer categories.[5] In this one-shot evaluation, higher accuracy also implies that the learned edit representation generalizes better from the specific edit pair to represent the semantics of an edit category.

Compared with GHE/Fixers-gold, the Fixers-one shot setting is somewhat more realistic, since in practice one could only provide similar edits on other input trees as the edit specifications. However, GHE/Fixers-gold provides a more *controllable* learning benchmark. It investigates how well an editor can perform when its given edit representation has encoded the exact desired edits on the given input. Note that this is not a trivial task as the edit representation has been "bottlenecked" within a single continuous vector. Particularly for GHE, it covers way more diverse edit patterns than the 16 fixer categories by Fixers, making the GHE-gold evaluation also challenging. Consequently, in our experiments, we examine each model by analyzing their performance on all evaluation settings.

**Baselines:** We compare our proposed neural editor (denoted as "**Graph2Edit**"[6]) with two state-of-the-art editors: (1) Graph2Tree (Yin et al., 2019), a model that, like ours, represents a program in its AST form and models the editing of tree-structured data. However, instead of generating a sequence of incremental edits, it decodes the edited tree in one pass; (2) CopySpan (Panthaplackel et al., 2020a), a model that represents programs as sequences of tokens and edits them by directly generating the edited code tokens from scratch.

We also experiment with two edit encoders for learning edit representations. Besides our proposed structural edit encoder (denoted as "**TreeDiff Edit Encoder**"), we consider a sequence edit encoder, which uses a bidirectional LSTM to compress three pieces of information: code tokens in $C_-$, code tokens in $C_+$, as well as their differences represented by a sequence of predefined edit tags (*e.g.* delete, add, or keep). This edit encoder (denoted as "Seq Edit Encoder") was shown to offer more precise and generalizable edit representations than others tested in Yin et al. (2019).

In experiments, we reproduce and test baselines by using implementations kindly provided by their authors. We include all configuration and implementation details in Appendix C.

## 5.2 Main Results

Tab. 1 shows our experimental results, where we examine two questions:

**(1) How does our incremental editor compare with one-pass baselines?** On Fixers-one shot, when all editors use the Seq Edit Encoder, our editor outperforms others substantially by more than 9% macro accuracy and 7% micro accuracy. This implies that our editor is better at capturing generalizable semantics underlying the edits. Given that all editors use the same architecture for the edit encoder, this also means that our editor encourages better edit repre-

Table 1: Test accuracy (%) of different editors and edit encoders. See more comparisons in § D.1.

| Model | GHE-gold | Fixers-gold | Fixers-one shot | |
| --- | --- | --- | --- | --- |
| | | | macro | micro |
| *w/ Seq Edit Encoder:* | | | | |
| CopySpan | 67.40 | 87.07 | 20.64 | 24.20 |
| Graph2Tree | 57.13 | 79.48 | 28.49 | 35.53 |
| Graph2Edit | 54.49 | 71.90 | **37.49** | **42.55** |
| *w/ TreeDiff Edit Encoder:* | | | | |
| Graph2Tree | 67.06 | 82.35 | 36.17 | 42.35 |
| Graph2Edit | **69.35** | **91.59** | 36.10 | 41.34 |

sentation learning in the edit encoder. The outstanding generalization ability of our editor demonstrates the advantage of modeling incremental edits; when our editor is trained to generate the edits rather than the edited tree from scratch, it implicitly drives its edit encoder to learn to capture the *salient* information about the *edits* (otherwise it has no means to generate the accurate edit sequence).

Intriguingly, we observe inverse performance from the three editors when their edit representation is or is not inferred from the gold edit pair; editors performing better on GHE/Fixers-gold (CopySpan

---

[5]Note that this one-shot evaluation procedure is different from the one used by Yin et al. (2019), so the results from Table 5 of Yin et al. (2019) are not comparable to ours. See § C.1 for details.

[6]"Graph" simply indicates the use of a graph neural network to encode a tree.

Table 2: Edited programs $C_+$ from each editor (w/ Seq Edit Encoder) given $\langle f_\Delta(C'_-, C'_+), C_-\rangle$ on Fixers. We show where our editor succeeds (Example 1) and fails (Example 2). More examples can be found in § D.2.

| | Example 1 | Example 2 |
|---|---|---|
| $\langle C'_-, C'_+\rangle$ | $C'_-$: AskNode(VAR0,new SelectString(VAR1.VAR2.ToString() ↪ +LITERAL)).ShouldBe(VAR1); 
 $C'_+$: AskNode(VAR0,new SelectString(VAR1.VAR2+LITERAL)). ↪ ShouldBe(VAR1); | $C'_-$: VAR0.VAR1=(int)VAR2.ColBegin; 
 $C'_+$: VAR0.VAR1=VAR2.ColBegin; |
| $\langle C_-, C_+\rangle$ | $C_-$: PersistAsync(VAR0,VAR1=>Sender.Tell(VAR1.VAR2. ↪ ToString()+LITERAL+VAR3.IncrementAndGet())); 
 $C_+$: PersistAsync(VAR0,VAR1=>Sender.Tell(VAR1.VAR2+ ↪ LITERAL+VAR3.IncrementAndGet())); | $C_-$: var VAR0=(Exception)VAR1. ↪ ExceptionFromProto(VAR2.Cause); 
 $C_+$: var VAR0=VAR1. ↪ ExceptionFromProto(VAR2.Cause); |
| CopySpan | $C_+$: PersistAsync(VAR0,VAR1=>Sender.Tell(VAR1.VAR2+ ↪ VAR3.IncrementAndGet())); | $C_+$: var VAR0=VAR1. ↪ ExceptionFromProto(VAR2.Cause); |
| Graph2Tree | $C_+$: PersistAsync(VAR0,VAR1=>VAR1.VAR2.ToString(VAR1. ↪ VAR2+LITERAL).IncrementAndGet()); | $C_+$: var VAR0=VAR2.Cause; |
| Graph2Edit | $C_+$: PersistAsync(VAR0,VAR1=>Sender.Tell(VAR1.VAR2+ ↪ LITERAL+VAR3.IncrementAndGet())); | $C_+$: var VAR0=VAR2.Cause; |

> Graph2Tree > Graph2Edit) consistently obtain worse accuracies on Fixers-one shot (CopySpan < Graph2Tree < Graph2Edit). We conjecture that when Seq Edit Encoder is jointly trained with the baseline editors, it tends to memorize the specific patterns about $C_+$ as opposed to the generalizable information about the edits when trained with our editor, because the baseline editors are trained to decode the exact content of $C_+$ from scratch. In comparison, this phenomenon is less prominent for Graph2Tree and more for CopySpan, since the former generates $C_+$ in the form of an *AST tree* while the latter generates $C_+$ in the form of a *token sequence* (which is exactly how $C_+$ is encoded by the Seq Edit Encoder).

**Case study & expressivity of structural edits:** We further showcase the generation from each editor (with Seq Edit Encoder) in the Fixers-one shot setting (Tab. 2). Example 1 illustrates typical cases where our editor Graph2Edit succeeds while the baseline editors fail. The example is about removing a redundant ToString call. Our editor learns to transfer and apply this editing pattern even when the input tree $C_-$ is very different from $C'_-$, while other editors behave very sensitively to the specific content of $C_-$. This is because, from the perspective of our editor, the edits required by $\langle C'_-, C'_+\rangle$ and $\langle C_-, C_+\rangle$ are the same, both first deleting an InvocationExpression subtree corresponding to "VAR1.VAR2.ToString()" and then copying back its MemberAccessExpression subtree corresponding to "VAR1.VAR2". In fact, we observe that in many cases, the actual tree edit that our editor needs to perform is irrelevant to the surface form of the input tree $C_-$. As our editor is trained to generate the actual tree edits, together with Seq Edit Encoder, it learns a better alignment between changing at the token level (*e.g.* from "VAR1.VAR2.ToString()" to "VAR1.VAR2") and performing targeted edits at the tree level.

On the other hand, this also means that our editor may fail when the desired edits for $\langle C_-, C_+\rangle$ bears a very different structure from the edits of $\langle C'_-, C'_+\rangle$, even if they are very close at the token level. Example 2 illustrates this situation where our editor fails. In this example, editing $C'_-$ involves removing a redundant type cast from a MemberAccessExpression subtree (corresponding to "VAR2.ColBegin") while the desired edits for $C_-$ require detaching the type cast from an InvocationExpression subtree (corresponding to "VAR1.ExceptionFromProto(VAR2.Cause)"). Therefore, even if our editor can precisely capture the structural edits expressed in $\langle C'_-, C'_+\rangle$, it cannot edit $C_-$ correctly. We observe that Graph2Tree could also be sensitive to this phenomenon while CopySpan runs successfully (although in many other cases, CopySpan produces ungrammatical programs, as we enumerate in § D.2).

Finally, we note that our editor also performs comparably with or better than Graph2Tree when they are both equipped with TreeDiff Edit Encoder, as we will discuss next.

**(2) What is the influence of edit encoding?** When replacing the Seq Edit Encoder with TreeDiff Edit Encoder, we observe significant improvement for both Graph2Tree and Graph2Edit on GHE/Fixers-gold; in the meantime, their performance on Fixers-one shot is comparable to the best accuracy. This implies that our proposed edit encoder is able to learn both more *expressive* and more *generalizable* edit representations. Particularly for our proposed Graph2Edit, it clearly outperforms Graph2Tree on GHE/Fixers-gold and is comparable to the latter on Fixers-one shot.

However, we also notice that for Graph2Edit, switching to the TreeDiff Edit Encoder only helps the GHE/Fixers-gold scenario and results in a slight performance drop in the one-shot setting. This is likely because Graph2Edit has overfit to the specific edit representations during training, the same issue that CopySpan confronts, when the target outputs of the editor (ground-truth tree edits for Graph2Edit and ground-truth code tokens for CopySpan) have been exposed to the edit encoder (TreeDiff Edit Encoder for Graph2Edit and Seq Edit Encoder for CopySpan) *in exactly the same format*. We note that the issue comes with the fact that all models are trained on the GHE-gold training set while only tested on Fixers-one shot (see § 3.3 and § 5.1). It would be ideal to *train* and *test* a model on the Fixers-one shot setting, but the Fixers dataset is not large enough for this to be feasible. We leave such an evaluation as an important topic to explore in the future.

Finally, in § D.3, we show the nearest neighbors of given edit pairs based on their edit representations, which qualitatively also demonstrate the superiority of TreeDiff Edit Encoder.

### 5.3 IMITATION LEARNING EXPERIMENTS

We finally demonstrate that training our editor via imitation learning makes it more robust. We consider two data settings: 20% or full training data. In each case, we first pretrain our editor with gold edit sequences on the training set via supervised learning, equivalent to setting $\beta$ to 1 in the first iteration of imitation learning, a commonly adopted strategy for DAGGER (Ross et al., 2011). We then run another iteration of imitation learning on the same training set to sample states and collect dynamic expert demonstrations, following either DAGGERSAMPLING (Algo. 1) or POSTREFINE-SAMPLING (Algo. 2). Empirically, we observe worse performance when setting $\beta = 0$ in DAGGER-SAMPLING. This is likely because in the editing tasks we experiment on, offering one-step expert demonstrations is not enough to teach the model to complete all the remaining edits successfully. We eventually set $\beta = 0.5$. We include the experimental details and an analysis in § D.4.

For the base editor, we use "Graph2Edit w/ Seq Edit Encoder," which is more prone to mistakes than "Graph2Edit w/ TreeDiff Edit Encoder" and thus presumably a better testbed for robust learning algorithms. The experimental results are show in Tab. 3. In the 20% training data setting, imitation learning improves supervised learning slightly with DAGGERSAMPLING and by 1.5% accuracy with POSTRE-FINESAMPLING. Our analysis shows that the editor trained using DAGGER-SAMPLING learns to correct its previous wrong edits (*e.g.* `Delete["VAR1"]` then `Add["StringX"]`). However, it may also fall into a local loop of repetitively deleting and

Table 3: Test (dev) accuracy by % of Graph2Edit w/ Seq Edit Encoder on GHE-gold after one iteration of imitation learning. Examples (simplified) illustrate how the base editor works when trained via supervised learning or with different imitation learning strategies. Colors indicate correct or incorrect edits.

|  | Supervised | DAGGER | POSTREFINE |
|---|---|---|---|
| w/ 20% data | 42.30 (44.22) | 42.70 (45.03) | **43.94** (**46.10**) |
| w/ full data | 54.49 (55.43) | 53.85 (55.57) | **54.91** (**56.48**) |
| Example | ... | ... | ... |
|  | Add[Id−>Token] | Add[Id−>Token] | Add[Id−>Token] |
|  | Add["VAR1"] | Add["VAR1"] | Add["StringX"] |
|  | ... | Delete["VAR1"] | ... |
|  |  | Add["StringX"] |  |
|  |  | ... |  |
| Avg. edits ($T$) | 7.480 | 11.549 | 7.594 |

adding the same component, which makes its edit length ($T$) generally longer than other editors. This situation is neatly remedied by using POSTREFINESAMPLING to collect expert demonstrations. With this strategy, although we train the editor to correct its wrong edits as a post refinement step, the well trained editor is indeed enhanced to be more robust in making correct decisions in its initial full pass of editing (rather than making wrong decisions then revoking them). This strategy also improves the base editor under the full training data setting slightly.

## 6 CONCLUSION AND FUTURE WORK

This paper presented a generic model for incremental editing of tree-structured data and demonstrated its capability using program editing as an example. In the future, this model could be extended to other tasks such syntax-based grammar error correction (Zhang & Wang, 2014) and sentence simplification (Feblowitz & Kauchak, 2013), or incorporate natural language-based edit specification, where the editing process is triggered by natural language feedback or commands (Suhr et al., 2018; Zhang et al., 2019; Yao et al., 2019; 2020; Elgohary et al., 2020).

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

## A MODEL ARCHITECTURE DETAILS

### A.1 IMPLEMENTATION WITH ASDL

To implement the "dummy node" mechanism, we utilize the ASDL "field", which ensures the grammatical correctness of every edit. In ASDL, children of each tree node are grouped under different fields, and each field has a cardinality property (single, optional ?, or sequential *) indicating the number of nodes they can accept as grammatically valid children.

For single-cardinality fields that require exactly one child and optional-cardinality fields that require optionally zero or one child, we attach one dummy node when they do not have a child. For example, at $t = 1$ in Fig. 1(b), when the `MethodCall`-rooted subtree is deleted, we automatically attach a `Dummy` node to its parent field "`right`" (which has single cardinality). Then the new node `ElementAccess` can be added by selecting `Dummy` and replacing it with the new node. Similarly, after deriving node `ElementAccess` at $t = 2$, we automatically add a `Dummy` node to each of its two single-cardinality fields (*i.e.* `obj` and `index`). Note that, to add a new tree node or copy a subtree as a child of a single/optional field, the `Add` or `CopySubTree` operator needs to be applied to a dummy node, because the dummy node of a single/optional field indicates the only vacant position that is syntactically valid to accept a child for such fields. Then the new tree node or the copied subtree will simply *replace* the original dummy node of the field.

For sequential-cardinality fields that accept multiple children, we always attach one dummy node as their rightmost child. For example, a sequential field having two children A and B will have a child list of $[A, B, Dummy]$. Adding a new node in this case is implemented by selecting the right sibling of the target position and then *inserting* the new node to its left. For example, adding a new node C to the left of A can then be achieved by selecting A and then inserting C before it, and adding to the right of B is done by selecting `Dummy` and then inserting C before `Dummy` (resulting in $[A, B, C, Dummy]$).

### A.2 TREE EDIT DECODER

In this section, we provide detailed formulations of node selection and value prediction in our proposed tree edit decoder.

**Node Selection:** Given a tree $g_t$, there could exist an arbitrary number of tree nodes. Therefore, we design the node selection module similar to a pointer network (Vinyals et al., 2015):

$$\boldsymbol{h}_{node,t} = \tanh(\boldsymbol{W}_{node}[\boldsymbol{s}_t; \text{emb}(op_t)] + \boldsymbol{b}_{node}),$$

$$p(n_t|\boldsymbol{s}_t, op_t) = \text{softmax}(\boldsymbol{h}_{node,t}^T \boldsymbol{n}_t),$$

where $\text{emb}(op_t)$ embeds the previously selected operator $op_t$, $\boldsymbol{n}_t$ is the node representation, and $\boldsymbol{W}_{node}, \boldsymbol{b}_{node}$ are model parameters. The softmax is computed over all nodes $n_t \in g_t$.

**Value Prediction:** After deciding the target position (inferred from the selected node), adding a new node or subtree to the current tree can be viewed as expanding its parent node in typical tree-based generation tasks (Yin & Neubig, 2017; Rabinovich et al., 2017; Yin & Neubig, 2018). We thus adapt the tree-based semantic parsing model of Yin & Neubig (2018) as our value predictor.

Recall that the `Add` operator adds a new node to the tree by either applying a production rule ($val = rule$) or predicting a terminal token ($val = tok$), and the `CopySubTree` operator copies a subtree ($val = subtree$) to expand the current tree. In all cases, we only consider candidates that satisfy the underlying grammar constraints. The prediction probability is also calculated via a pointer network in order to handle varying numbers of valid candidates in each decision situation:

$$\boldsymbol{h}_{val,t} = \tanh(\boldsymbol{W}_{val}[\boldsymbol{s}_t; \boldsymbol{n}_t; \text{emb}(p_{n_t} \mapsto n_t)] + \boldsymbol{b}_{val}),$$

$$p(val_t|\boldsymbol{s}_t, op_t, n_t) = \text{softmax}(\boldsymbol{h}_{val,t}^T \boldsymbol{W} \text{emb}(val_t)),$$

where $\boldsymbol{W}_{val}, \boldsymbol{b}_{val}$ and $\boldsymbol{W}$ are all model parameters, $\text{emb}(p_{n_t} \mapsto n_t)$ is the embedding of the edge type (or "field"; see § A.1) between the parent node $p_{n_t}$ and the child $n_t$ (*e.g.* field "right" for `AssignStmt` $\xrightarrow{\text{right}}$ `ElementAccess`), and $\text{emb}(val_t)$ denotes the representation of the argument candidate: for production rules, it is their learned embedding; for terminal tokens, it is their word embedding; for subtree candidates, we use the representation of their root node as the encoding of the subtree.

---

**Algorithm 1** DAGGERSAMPLING

**Require:** $\langle f_\Delta, C_-, C_+ \rangle$ from training set, learning editor $\pi_\theta$, expert policy $\pi^*$, $\beta \in [0,1]$
1: Let $g_1 = C_-$.
2: Let $\pi' = \beta\pi^* + (1-\beta)\pi_\theta$.
3: Sample a trajectory from $\pi'(f_\Delta, g_1)$.
4: Collect and return $\{\langle s, \pi^*(s)\rangle\}$ for all states $s$ visited by $\pi'$.

---

---

**Algorithm 2** POSTREFINESAMPLING

**Require:** $\langle f_\Delta, C_-, C_+ \rangle$ from training set, learning editor $\pi_\theta$, expert policy $\pi^*$
1: Let $g_1 = C_-$.
2: Sample a trajectory using $\pi_\theta(f_\Delta, g_1)$. Denote $g_T$ as the output tree by the editor.
3: **if** $g_T \neq C_+$ **then**
4:     Sample a trajectory from $\pi^*(f_\Delta, g_T)$;
5:     Return $\{\langle s_t, \pi^*(s_t)\rangle | t \geq T\}$.
6: **else**
7:     Return empty collection.
8: **end if**

---

## B   IMITATION LEARNING ALGORITHMS

We present DAGGERSAMPLING and POSTREFINESAMPLING in Algo. 1 and Algo. 2, respectively.

## C   DATASETS AND CONFIGURATIONS

### C.1   DATASETS

For all datasets, we use the preprocessed version by Yin et al. (2019) for a fair comparison. The preprocessing includes tokenizing each code snippet and converting it into a AST.[7] For each $\langle C_-, C_+ \rangle$, we run a dynamic programming algorithm to search for the shortest edit sequence from $C_-$ to $C_+$. The average length of gold edit sequences is 7.375 on GitHubEdits training set and 7.348 on C#Fixers.

Our evaluation of the Fixers-one shot setting follows Panthaplackel et al. (2020a). Specifically, for each fixer category, we pick its first 100 samples to compute 100 "seed" edit representations (for categories whose numbers of samples are smaller than 100, we use all of their $N$ samples). We then apply each seed edit representation to edit the remaining 99 samples (or $N$-1 samples for categories with less than 100 total samples), calculate one accuracy score for each seed, and report an average accuracy over the 100 seeds. This gives us an average score for each fixer category. Because the sample size of each fixer category is highly imbalanced, we report both *macro average* (treating all categories equally) and *micro average* (dependent on sample size) edit accuracies over the 16 fixer categories. Note that this is different from the evaluation procedure of Yin et al. (2019). Yin et al. (2019) considered only 10 "seeds" per category (although each seed edit representation is applied to all samples in the category), and reported the *best* accuracy over the 10 seeds as the score for the category. We believe enlarging the number of "seeds" and reporting an *average* accuracy can better represent a model's capability. As a result, the numbers in Table 5 of Yin et al. (2019) are not comparable with results reported in our Tab. 1.

As introduced in § 3.3, for every $\langle C_-, C_+ \rangle$ pair in the training set, we run a dynamic programming algorithm to create the gold-standard edit action sequence. We elaborate the algorithm in Algo. 3 (for simplicity, we omit the edit backtrace recording part). The algorithm edits a source tree node $C^s$ into a target tree node $C^t$ of the same type (and thus having the same fields), given a memory of subtrees $\mathbb{M}$ that can be copied during edits. In practice, this subtree memory consists of all subtrees in the input tree $C_-$. The algorithm compares the source and the target tree node field by field (Line 2). In each field $f$, we assume a left-to-right, top-down order when comparing the children of the source tree node $C^{s,f}$ and the children of the target tree node $C^{t,f}$ within this field. $C_m^{s,f}$ (resp. $C_n^{t,f}$) denotes the $m$-th ($n$-th) child of $C^{s,f}$ ($C^{t,f}$). "CountTreeNode" counts the number of tree nodes within the subtree of a root node.

$\boldsymbol{D}[m,n]$ defines the shortest distance of editing $C^{s,f}$ (up to the $m$-th child) into $C^{t,f}$ (up to the $n$-th child). In Line 4-19, the algorithm initializes the distance matrix with boundary cases; in Line

---

[7]The ASDL grammar we used for C# can be found at: `https://raw.githubusercontent.com/dotnet/roslyn/master/src/Compilers/CSharp/Portable/Syntax/Syntax.xml`.

---

**Algorithm 3** TREESHORTESTDIST

---

**Require:** Source tree node $C^s$ and target tree node $C^t$ (assuming non-terminal nodes and have the same node type and fields), subtree memory $\mathbb{M}$.

**Ensure:**

1: Initialize $d \leftarrow 0$; // total edit distance for all fields
2: **for** every field $f$ of $C^s$ **do** // loop over aligned fields
3:     $M \leftarrow$ number of children in $C^{s,f}$, $N \leftarrow$ number of children in $C^{t,f}$.
4:     Initialize $\boldsymbol{D}$ as a matrix of $(M + 1)$ by $(N + 1)$. // distance matrix
5:     Initialize $\boldsymbol{D}[0,0] \leftarrow 0$.
6:     **for** $m = 1$ to $M$ **do**
7:         $\boldsymbol{D}[m,0] \leftarrow \boldsymbol{D}[m-1,0]$ if $C^{s,f}_m$ is dummy and $\boldsymbol{D}[m-1,0]+1$ otherwise. //deleting a tree node takes 1 step; no need to delete dummy nodes
8:     **end for**
9:     **for** $n = 1$ to $N$ **do**
10:         **if** field $f$ is single/optional and $C^{s,f}_m$ is a valid, non-dummy node **then**
11:             $\boldsymbol{D}[0,n] \leftarrow$ inf. //cannot add to non-empty single/optional fields of $C^{s,f}$
12:         **else if** $C^{t,f}_n$ is dummy **then**
13:             $\boldsymbol{D}[0,n] \leftarrow \boldsymbol{D}[0,n-1]$. //no need to add dummy nodes
14:         **else if** $C^{t,f}_n \in \mathbb{M}$ **then**
15:             $\boldsymbol{D}[0,n] \leftarrow \boldsymbol{D}[0,n-1]+1$. // copying a subtree takes 1 step
16:         **else**
17:             $\boldsymbol{D}[0,n] \leftarrow \boldsymbol{D}[0,n-1] + \text{CountTreeNode}(C^{t,f}_n)$. // add nodes of $C^{t,f}_n$ one by one, each requiring 1 step
18:         **end if**
19:     **end for**
20:     **for** $m = 1$ to $M$ **do**
21:         **for** $n = 1$ to $N$ **do**
22:             $v_1 \leftarrow \boldsymbol{D}[m-1,n]$ if $C^{s,f}_m$ is dummy and $\boldsymbol{D}[m-1,n]+1$ otherwise;
23:             $v_2 \leftarrow \boldsymbol{D}[m,n-1]$ if $C^{t,f}_n$ is dummy, $\boldsymbol{D}[m,n-1]+1$ if $C^{t,f}_n \in \mathbb{M}$, and $\boldsymbol{D}[m,n-1] + \text{CountTreeNode}(C^{t,f}_n)$ otherwise;
24:             $v_3 \leftarrow \boldsymbol{D}[m-1,n-1] + \text{TREESHORTESTDIST}(C^{s,f}_m, C^{t,f}_n, \mathbb{M})$ if $C^{s,f}_m$ and $C^{t,f}_n$ are both non-terminal nodes and have the same node type, $\boldsymbol{D}[m-1,n-1]$ if $C^{s,f}_m$ and $C^{t,f}_n$ are both terminal tokens and have the same value, and inf otherwise;
25:             $\boldsymbol{D}[m,n] \leftarrow \min\{v_1, v_2, v_3\}$.
26:         **end for**
27:     **end for**
28:     $d \leftarrow d + \boldsymbol{D}[M,N]$. // add the shortest distance of field $f$
29: **end for**
30: **return** $d$ as the total shortest edit distance.

---

20-27, it considers general cases. $\boldsymbol{D}[M,N]$ is thus the shortest distance of editing $C^{s,f}$ into $C^{t,f}$ completely (Line 28). The final distance from $C^s$ to $C^t$ is the summation of shortest distances over all fields (Line 30).

To generate the shortest edit distance for $\langle C_-, C_+ \rangle$, we run the TREESHORTESTDIST algorithm with $C^s$ and $C^t$ being the roots of $C_-$ and $C_+$, respectively. Based on the backtrace records (omitted in Algo. 3), we produce the gold-standard edit action sequence, with one extra `Stop` edit action appended in the end to signal the end of the editing process. Note that this is a global stop signal. As the model learns about when to stop editing the whole tree globally, it actually also learns about when to stop editing any certain subtrees within it.

### C.2  MODEL CONFIGURATIONS

Since surrounding contexts around the edited program are also provided in all datasets, we additionally allow the value predictor (§ 3.1) to copy a terminal token from either the input tree's code tokens or the contexts. To this end, we introduce another bidirectional LSTM encoder to encode the input code tokens as well as the contexts. The last hidden state is used to represent each token. The same design is also adopted in the two baseline editors.

For the encoder of our neural editor, the dimension of word embedding and the tree node representation is set to 128. The dimension of the bidirectional LSTM encoder for encoding input code tokens and contexts is set to 64. The hidden state for tracking tree history is set to 256 dimensions. In the decoder side, the dimensions of the operator embedding, the field embedding, the production rule embedding, and the hidden vector in value prediction are set to 32, 32, 128 and 256, respectively.

For a fair comparison, we follow Yin et al. (2019) and Panthaplackel et al. (2020a) to encode a code edit into a real-valued vector of 512 dimensions. For our TreeDiff Edit Encoder, each edit action is encoded into a vector of 256 dimensions. The bidirectional LSTM also has a hidden state of 256 dimensions. When training Graph2Edit/Graph2Tree jointly with TreeDiff Edit Encoder, common parameters that are designed for both the neural editor and the edit encoder (*e.g.* the operator/field embedding) are shared.

In experiments, we reproduce and evaluate baselines by using implementations kindly provided by their authors. This includes testing the baseline editors under exactly the same setting as they were tested in their original paper (*e.g.* decoding using beam search of size 5 for Graph2Tree and 20 for CopySpan).

For the supervised learning, we train our Graph2Edit for 30 epochs on GitHubEdits training set, where the best model parameters are selected based on the editor's cross entropy loss on dev set. To enable more stable and reproducible results, we repeat the experiments for 3 times and report average performance.

## D  ADDITIONAL EXPERIMENTAL RESULTS

### D.1  A MORE COMPREHENSIVE COMPARISON WITH EXISTING APPROACHES

We note that there are some other approaches by Yin et al. (2019) and Panthaplackel et al. (2020a) that are not included in our Tab. 1. However, these approaches have been reported to be weaker than those we mainly tested. For example, "Seq2Seq + Seq Edit" performs worse than "CopySpan + Seq Edit" on both GHE-gold and Fixers-one shot (micro) in Panthaplackel et al. (2020a); Yin et al. (2019) claimed that "Graph2Tree + Seq Edit" is better than both "Seq2Seq + Seq Edit" and "Graph2Tree + Graph Edit". As we have slightly adjusted the evaluation settings (*i.e.* adding Fixers-gold and changing the evaluation procedure of Fixers-one shot compared with Yin et al. (2019)), results are largely missing for existing approaches. Therefore, we only re-tested and compared with existing state-of-the-art approaches (CopySpan and Graph2Tree) in our main results (§ 5.2). In Tab. 4, however, we include a more comprehensive set of existing approaches for reference.

Table 4: A more comprehensive comparison with existing approaches. [*]: numbers copied from original paper; –: missing evaluation in the original paper; other numbers without indication are reported by us. For baselines re-tested by us, we use implementations kindly provided by their authors and make sure that our reproduced version has comparable performance as what was reported in their paper.

| Model | GHE-gold | Fixers-gold | Fixers-one shot | |
| --- | --- | --- | --- | --- |
| | | | macro | micro |
| Seq2Seq + Seq Edit (Yin et al., 2019) | 59.63[*] | – | – | – |
| Seq2Seq + Seq Edit (Re-implemented w/ tricks by Panthaplackel et al. (2020a)) | 64.40[*] | – | – | 18.80[*] |
| CopySpan + Seq Edit (Panthaplackel et al., 2020a) | 67.40[*] | 87.07 | 20.64 | 24.20[*] |
| Graph2Tree + Seq Edit (Yin et al., 2019) | 57.13 | 79.48 | 28.49 | 35.53 |
| Graph2Edit + Seq Edit (ours) | 54.49 | 71.90 | **37.49** | **42.55** |
| Graph2Tree + Graph Edit (Yin et al., 2019) | 48.05[*] | – | – | – |
| Graph2Tree + TreeDiff Edit (w/ our edit encoder) | 67.06 | 82.53 | 36.17 | 42.35 |
| Graph2Edit + TreeDiff Edit (ours) | **69.35** | **91.59** | 36.10 | 41.34 |

## D.2 MORE EDIT EXAMPLES

In Tab. 5, we include more examples from each editor (with Seq Edit Encoder) in the Fixers-one shot setting. Example 1-2 demonstrate that our proposed Graph2Edit editor can successfully identify the correct target position to edit, even when the context of $C_-$ is very different from that of $C'_-$. Consider Example 1 for instance. Graph2Edit correctly locates the `await` keyword and its associated expression "`await VAR0.GracefulStop(TimeSpan.FromSeconds(LITERAL))`", and then appends "`.ConfigureAwait(false)`" to the expression, all within the argument field of the outer "`Assert.True()`" expression. Graph2Tree also tries to append "`.ConfigureAwait(false)`" to the `await` expression, but as it generates the edited tree from scratch, it mistakenly copies an additional argument to the "`Assert.True()`" expression. CopySpan's generation misses a right parenthesis, leading to an ungrammatical program. Similarly in Example 2, only Graph2Edit performs the desired edits when the contexts are different for $C'_-$ and $C_-$.

On the other hand, Graph2Edit can fail when the correct edits require more than structural information. For example, to succeed in Example 3 of Tab. 5, an editor needs to generalize the removal of redundant parentheses from a literal variable name ("VAR4") to a bracket-wrapped binary expression ("VAR2/LITERAL"). We found that all editors fail in this case. Graph2Edit in this example correctly removes the parenthesized binary expression but can only copy the literal part ("LITERAL") back; Graph2Tree incorrectly revises the irrelevant VAR3 variable; CopySpan again produces an ungrammatical output. The target edits in Example 4 are even more complicated.[8] It requires an editor to understand the use of the `nameof` operator to replace the explicit string literal. All editors fail to identify "`EnumUnderTest.ALLCAPITALS.ToString()`" as another kind of string literal.

---

[8] A de-anonymized instance of $\langle C'_-, C'_+ \rangle$ could be: $\langle$`parameter.EnsureIsPositiveFinite("parameter")`, `parameter.EnsureIsPositiveFinite(nameof(parameter))`$\rangle$. The `nameof` operator returns the name of the variable as a string literal.

Table 5: Edited programs $C_+$ from each editor (w/ Seq Edit Encoder) given $\langle f_\Delta(C'_-, C'_+), C_-\rangle$ on Fixers. We show where our editor succeeds in Example 1-2 and fails in Example 3-4.

| | Example 1 |
|---|---|
| $\langle C'_-, C'_+\rangle$ | $C'_-$: `await VAR0.WriteAsync(VAR1.Array, VAR1.Offset, VAR1.VAR2);`
$C'_+$: `await VAR0.WriteAsync(VAR1.Array, VAR1.Offset, VAR1.VAR2).ConfigureAwait(false);` |
| $\langle C_-, C_+\rangle$ | $C_-$: `Assert.True(await VAR0.GracefulStop(TimeSpan.FromSeconds(LITERAL)));`
$C_+$: `Assert.True(await VAR0.GracefulStop(TimeSpan.FromSeconds(LITERAL)).`
↪ `ConfigureAwait(false));` |
| CopySpan | $C_+$: `Assert.True(await VAR0.GracefulStop(TimeSpan.FromSeconds(LITERAL)).`
↪ `ConfigureAwait(false);` |
| Graph2Tree | $C_+$: `Assert.True(await VAR0.GracefulStop(TimeSpan.FromSeconds(LITERAL)), await VAR0.`
↪ `GracefulStop(TimeSpan.FromSeconds(LITERAL)).ConfigureAwait(false));` |
| Graph2Edit | $C_+$: `Assert.True(await VAR0.GracefulStop(TimeSpan.FromSeconds(LITERAL)).`
↪ `ConfigureAwait(false));` |
| | Example 2 |
| $\langle C'_-, C'_+\rangle$ | $C'_-$: `VAR0 = (VAR0*LITERAL)^(VAR1 != null ? VAR1.GetHashCode(): 0);`
$C'_+$: `VAR0 = (VAR0*LITERAL)^(VAR1?.GetHashCode()?? 0);` |
| $\langle C_-, C_+\rangle$ | $C_-$: `var VAR0 = (VAR1 != null ? VAR1.GetHashCode(): 0);`
$C_+$: `var VAR0 = (VAR1?.GetHashCode()?? 0);` |
| CopySpan | $C_+$: `var VAR0 = (VAR1 != null ? VAR1?.GetHashCode()?? 0);` |
| Graph2Tree | $C_+$: `var VAR0 = (VAR1 != null ? VAR1.GetHashCode(): 0);` |
| Graph2Edit | $C_+$: `var VAR0 = (VAR1?.GetHashCode()?? 0);` |
| | Example 3 |
| $\langle C'_-, C'_+\rangle$ | $C'_-$: `Push(VAR0.VAR1, VAR0.VAR2(VAR0.VAR3.Select(VAR4 => Grab((VAR4))).`
↪ `ToImmutableList()));`
$C'_+$: `Push(VAR0.VAR1, VAR0.VAR2(VAR0.VAR3.Select(VAR4 => Grab(VAR4)).ToImmutableList`
↪ `()));` |
| $\langle C_-, C_+\rangle$ | $C_-$: `VAR0.Add(ApplyGender(VAR1[(VAR2/LITERAL)], VAR3));`
$C_+$: `VAR0.Add(ApplyGender(VAR1[VAR2/LITERAL], VAR3));` |
| CopySpan | $C_+$: `VAR0.Add(ApplyGender(VAR1[(VAR2/LITERAL[(/LITERAL/(LITERAL)], VAR3));` |
| Graph2Tree | $C_+$: `VAR0.Add(ApplyGender(VAR1[(VAR2/LITERAL)], (VAR2/VAR2.VAR3).VAR3(VAR3)));` |
| Graph2Edit | $C_+$: `VAR0.Add(ApplyGender(VAR1[LITERAL], VAR3));` |
| | Example 4 |
| $\langle C'_-, C'_+\rangle$ | $C'_-$: `VAR0.EnsureIsPositiveFinite(LITERAL);`
$C'_+$: `VAR0.EnsureIsPositiveFinite(nameof(VAR0));` |
| $\langle C_-, C_+\rangle$ | $C_-$: `Assert.Equal(EnumUnderTest.ALLCAPITALS.ToString(), EnumUnderTest.ALLCAPITALS.`
↪ `Humanize());`
$C_+$: `Assert.Equal(nameof(EnumUnderTest.ALLCAPITALS), EnumUnderTest.ALLCAPITALS.`
↪ `Humanize());` |
| CopySpan | $C_+$: `Assert.Equal(nameof(VAR1)));` |
| Graph2Tree | $C_+$: `Assert.Equal(nameof(VAR1));` |
| Graph2Edit | $C_+$: `Assert.Equal(EnumUnderTest.ALLCAPITALS.ToString(), EnumUnderTest.ALLCAPITALS.`
↪ `CSharpName(VAR1));` |

## D.3 EDIT REPRESENTATIONS OF TREEDIFF EDIT ENCODER

Tab. 6 shows the nearest neighbors of given edit pairs from GHE dev set, based on the cosine similarity of their edit representations $f_\Delta(C_-, C_+)$ calculated by different edit encoders. The edit in Example 1 means to swap function arguments (*e.g.* from "`(VAR0,VAR1)`" to "`(VAR1, VAR0)`"). Intuitively such structural changes can be easily captured by our tree-level edit encoder. This is consistent with our results, which show that, for both Graph2Tree and Graph2Edit, TreeDiff Edit Encoder learns more consistent edit representations for this edit, while Seq Edit Encoder may confuse it with edits that replace the original argument with a new one (*e.g.* modifying "`(VAR0,VAR1)`" to "`(VAR2,VAR0)`"). Our proposed edit encoder can also generalize from literals (*e.g.* swapping between "`(VAR0,VAR1)`") to more complex expressions (*e.g.* swapping between "`(VAR0.Value, LITERAL)`"). On the other hand, when the intended edits can be easily expressed as token-level

Table 6: The nearest neighbors of given edit pairs based on their edit representations.

| Example 1 | Example 2 |
|---|---|

**Example 1**

$C_-$: `BoundsCheck(VAR0, VAR1);`
$C_+$: `BoundsCheck(VAR1, VAR0);`

Graph2Tree – Seq Edit Encoder
▶ $C_-$: `ReleasePooledConnectorInternal(VAR0, VAR1);`
  $C_+$: `ReleasePooledConnectorInternal(VAR2, VAR0);`

▶ $C_-$: `UngetPooledConnector(VAR0, VAR1);`
  $C_+$: `UngetPooledConnector(VAR2, VAR0);`

▶ $C_-$: `VAR0.Warn(LITERAL, VAR1);`
  $C_+$: `VAR0.Warn(VAR1, LITERAL);`

Graph2Tree – TreeDiff Edit Encoder
▶ $C_-$: `InternalLogger.Error(LITERAL, VAR0);`
  $C_+$: `InternalLogger.Error(VAR0, LITERAL);`

▶ $C_-$: `VAR0.Warn(LITERAL, VAR1);`
  $C_+$: `VAR0.Warn(VAR1, LITERAL);`

▶ $C_-$: `AssertEqual(VAR0.Value, LITERAL);`
  $C_+$: `AssertEqual(LITERAL, VAR0.Value);`

Graph2Edit – Seq Edit Encoder
▶ $C_-$: `ReleasePooledConnectorInternal(VAR0, VAR1);`
  $C_+$: `ReleasePooledConnectorInternal(VAR2, VAR0);`

▶ $C_-$: `UngetPooledConnector(VAR0, VAR1);`
  $C_+$: `UngetPooledConnector(VAR2, VAR0);`

▶ $C_-$: `ReportUnusedImports(VAR0, VAR1, VAR2);`
  $C_+$: `ReportUnusedImports(VAR2, VAR0, VAR1);`

Graph2Edit – TreeDiff Edit Encoder
▶ $C_-$: `VAR0.Warn(LITERAL, VAR1);`
  $C_+$: `VAR0.Warn(VAR1, LITERAL);`

▶ $C_-$: `InternalLogger.Error(LITERAL, VAR0);`
  $C_+$: `InternalLogger.Error(VAR0, LITERAL);`

▶ $C_-$: `AssertEqual(VAR0.Value, LITERAL);`
  $C_+$: `AssertEqual(LITERAL, VAR0.Value);`

**Example 2**

$C_-$: `var VAR0=GetEtagFromRequest();`
$C_+$: `var VAR0=GetLongFromHeaders(LITERAL);`

Graph2Tree – Seq Edit Encoder
▶ $C_-$: `var VAR0=new ProfileConfiguration();`
  $C_+$: `var VAR0=new Profile(LITERAL);`

▶ $C_-$: `var VAR0=PrepareForSaveChanges();`
  $C_+$: `var VAR0=PrepareForSaveChanges(null);`

▶ $C_-$: `bool VAR0=true;`
  $C_+$: `bool VAR0=CanBeNull(VAR1);`

Graph2Tree – TreeDiff Edit Encoder
▶ $C_-$: `var VAR0=new ProfileConfiguration();`
  $C_+$: `var VAR0=new Profile(LITERAL);`

▶ $C_-$: `CalcGridAreas();`
  $C_+$: `SetDataSource(VAR0, VAR1);`

▶ $C_-$: `VAR0=new Win32PageFileBackedMemoryMappedPager();`
  $C_+$: `VAR0=new Win32PageFileBackedMemoryMappedPager(LITERAL);`

Graph2Edit – Seq Edit Encoder
▶ $C_-$: `var VAR0=new ProfileConfiguration();`
  $C_+$: `var VAR0=new Profile(LITERAL);`

▶ $C_-$: `VAR0.Dispose();`
  $C_+$: `VAR0.Close(VAR1);`

▶ $C_-$: `VAR0=VAR1(VAR2);`
  $C_+$: `VAR0=GetSpans(VAR2, VAR1);`

Graph2Edit – TreeDiff Edit Encoder
▶ $C_-$: `var VAR0=new ProfileConfiguration();`
  $C_+$: `var VAR0=new Profile(LITERAL);`

▶ $C_-$: `VAR0=Thread.GetDomain().DefineDynamicAssembly(VAR1, ↪ AssemblyBuilderAccess.Run);`
  $C_+$: `VAR0=Thread.GetDomain().DefineDynamicAssembly(VAR1, ↪ AssemblyBuilderAccess.RunAndSave, LITERAL);`

▶ $C_-$: `new DocumentsCrud().EtagsArePersistedWithDeletes();`
  $C_+$: `new DocumentsCrud().PutAndGetDocumentById(LITERAL);`

editing (*e.g.* inserting an argument token), the two edit encoders perform comparably, as shown in Example 2. However, we still observe that TreeDiff Edit Encoder works better at interpreting the editing semantics of code snippets with complex structures (*e.g.* more complex edit pairs are retrieved).

## D.4 MORE DETAILS ABOUT IMITATION LEARNING EXPERIMENTS

**Experimental Setup** We use "Graph2Edit w/ Seq Edit Encoder" as the base editor. We do not experiment with "Graph2Edit w/ TreeDiff Edit Encoder" as it performed very well on GHE-gold training set even without imitation learning. In fact, 80% of the remaining errors were due to issues such as unknown tokens, which cannot be fixed with better training algorithms, as they are outside the search space of our current model. We leave experimenting this model on harder datasets as future work. Like the main experiments (§ C.2), we ran the imitation learning experiments for three times and reported average performance in Tab. 3.

**Analysis** We provide a more detailed analysis about the imitation learning experiments, especially how the choice of $\beta$ in DAGGERSAMPLING affects the model performance. Our analysis is based on Graph2Edit+Seq Edit Encoder's results on the GHE dev set, when they are trained with 20% of the GHE training data. We observe that the DAGGERSAMPLING algorithm generally trains the editor to behave very *unstably*. The editor shows to "regret" its previous decisions (mostly about terminal tokens, the prediction of which is generally harder than that of non-terminal nodes on GHE). An example is shown in Tab. 3, where the DAGGERSAMPLING editor first adds a token "VAR1" to the tree and then deletes it in the next step. This happens to around 23% of the examples (count=2,373 in Tab. 7) on the dev set for DAGGERSAMPLING when $\beta$ is set to 0 (*i.e.* when the editor samples all states from itself throughout the imitation learning process). Among them, 84% of the deletions (count=1,989) are correct; they remove the a wrong token added in the previous step. However, we

also observe that the editor may then regret their correct deletions and add back the wrong tokens after "swinging" for a few steps (42% of the cases), revealing an interesting "add - delete (- add - ... - delete) - add" hesitation phenomenon. Similarly, among the remaining 384 examples where the editor deletes a correct token that it previously added, we found out in 51% of the cases the editor will add back the correct token. This unstable behavior can easily lead to an endless loop of repetitively adding and then deleting the same token (447 cases), until the editor reaches the maximum edit length that we set as a hyper-parameter (70 in our experiments).

Table 7: Analysis of DAGGERSAMPLING algorithms with $\beta$ being 0 and 0.5 on dev set. We analyze each algorithm's behavior about "adding then deleting" the same token.

|  | $\beta = 0$ | $\beta = 0.5$ |
|---|---|---|
| #of examples with add-then-delete | 2,373 | 1,032 |
| - #of endless edit loop | 447 | 605 |
| #of add wrong token then delete | 1,989 | 861 |
| - #of then add back | 845 | 467 |
| #of add correct token then delete | 384 | 171 |
| - #of then add back | 196 | 71 |

We hypothesize that setting $\beta$ to 0 in the DAGGERSAMPLING algorithm has forced the editor to learn only *single-step* correction edits (i.e., the correction edit under the current state). In other words, even if the editor could imitate the expert demonstration and correct its mistake at this single step, it still does not know how to proceed correctly for the next step. This is possibly the reason why the editor swings between adding then deleting the same token. An interesting question is why this problem shows more severely in our setting, compared with traditional imitation learning tasks (*e.g.* racing games). We hypothesize that editing tree-structured data for program revision requires more *continuous* and *structured* supervision, such as teaching the policy to complete the entire generation of a subtree, rather than teaching it to add only the next correct single tree node. We leave a deeper study of this problem to the future.

In experiments, to alleviate this problem, we propose to set $\beta$ higher, because when the editor executes actions from both itself and the expert, it is more likely to collect a continuous sequence of correction edits. We show improved program editing accuracy of DAGGERSAMPLING with $\beta = 0.5$ in Tab. 3, and an analysis of its "add-then-delete" behavior in Tab. 7. Compared with when $\beta = 0$, this setting reduces the frequency of unstable editing. We also propose a new sampling algorithm, POSTREFINESAMPLING. We observe that the POSTREFINESAMPLING algorithm trains the editor to perform much more stably than the DAGGERSAMPLING algorithm. We observe almost no (less than 0.5% of dev examples) the aforementioned add-then-delete behavior.

