# OpenReview forum: "Learning Structural Edits via Incremental Tree Transformations"
_ICLR.cc/2021/Conference — ICLR 2021 Poster_

### Official Review · AnonReviewer1 · 2020-10-14
**Interesting paper with strong empirical results**

**Rating:** 8
**Confidence:** 4

**Review:**

### Summary ###
The paper presents an approach for predicting edits in programs, by modeling the programs as trees. The approach is mainly an extension of Yin et al. (2019), with the main difference that the model is required to predict only the output **actions**, instead of generating the entire output tree as in Yin et al. (2019). This difference of predicting only output actions is shared with other previous work though.
The most interesting part in my opinion is the "imitation learning" improvement: during training, the model is trained to correct its own mistakes by "imitating" an expert that fixes the incorrect predictions.

Overall, I vote for acceptance. Although the technical contribution is limited, the paper presents strong empirical results and a combination of interesting ideas. I think that the paper could be easily further improved, as detailed below.

### Strengths ###
1. The paper presents improved results over the Graph2Tree model (Yin et al. 2019) and over CopySpan (Panthaplackel et al., 2020a).

2. The imitation learning part is very interesting, and its applicability for programs is novel as far as I know. I feel like maybe this should have been the main focus of the paper.

### Weaknesses ###
1. Limited novelty - the encoder, as far as I understand, is identical to the edit encoder of Graph2Tree (Yin et al. 2019). The decoder ("editor") is better, empirically and conceptually, than the decoder of Graph2Tree, but its main novelty is the prediction of the edit action itself, rather than generating the entire output tree. To me, this idea is not novel, as it was used in Tarlow et al. (2019), Dinella et al. (2020), and Brody et al. (2020).

2. A conceptual comparison with previous work is missing. First, the work of Tarlow et al. (2019) is not cited at all (although their application is different, the approach is very similar). Second, a comparison to Hoppity (Dinella et al., 2020) and to C3PO (Brody et al., 2020) is presented in a single paragraph and contains the following arguments:

(a) “While some recent works have examined models that make changes to trees for specific applications such as program bug fixing (Dinella et al., 2020) or edit completion (Brody et al., 2020), our method is designed to be generic and flexible in nature”
-- I don't think that this is a fair argument. These previous works are as general as this paper, they were just demonstrated on slightly different datasets.

(b) “it supports general tree edits including adding new tree nodes or copying a subtree, which are not fully allowed by previous work”
-- Can't Hoppity add new tree nodes? Can you clarify the classes of edits that previous works could not express and that this paper can express?

(c) “all tree edit operations are language-agnostic owing to the adoption of Abstract Syntax Description Language (ASDL; Wang et al. (1997)), which allows us to process arbitrary tree-based languages”
-- I don't think that this is a fair argument. These previous works are language-agnostic as well, they just use a different AST "format".

(d) “unlike the short edit sequences handled in previous work (e.g. up to three edits in Dinella et al. (2020)), we demonstrate our method’s applicability to much longer edit sequences”
-- The fact that Hoppity was evaluated with 3 edit actions does not mean that it is not applicable for longer action sequences. This argument would have been valid if it was demonstrated empirically that Hoppity's accuracy decreases as the length of the sequence increases.  In that case, the imitation learning part might be a very natural fix (which was good! but not shown).


3. Evaluation - it is unclear which datasets should we _really_ care about, and which are the main results. It seems that the proposed Graph2Edit model outperforms the Graph2Tree model (Yin et al. 2019) only in the "gold" datasets (GHE-gold and Fixers-gold), which expose (indirectly) the labels to the input. That is, these serve as "intrinsic" tasks that cannot really be compared across models. As far as I understand, Yin et al. (2019) argued that the "Fixers-one shot" is the dataset that really matters, and that GHE-gold and Fixers-gold are just "intermediary"/"intrinsic" training objectives.
In the "intermediary" datasets, Graph2Edit outperforms Graph2Tree, but it is not compared to "Seq2seq encoder+editor" which performed best in these datasets in Yin et al., 2019 (Table 4 in Yin et al., 2019).
In the one-shot dataset (Fixers one-shot) - Graph2Tree performs better than the proposed Graph2Edit model.
So, I am not sure what are the main results that the readers should focus on and what is the correct baseline. See question 1 below.

### Questions for Authors ###
1. Is the accuracy on the "gold" datasets (GHE-gold and Fixers-gold) really meaningful? Isn't this accuracy just an intermediary accuracy?
As far as I understand, when trained on these gold datasets: $f_{\Delta}$ depend on $C_{+}$, and then $f_{\Delta}$ is *used* in the prediction of $C_{+}$. I saw the footnote that says that  $f_{\Delta}$ does not *directly* expose $C_{+}$. So it means that it *indirectly* expose $C_{+}$, right? Section 3.3 explicitly says that "given an input tree $C_{-}$ and an edit representation  $f_{\Delta}$ (calculated either from $<C_{-}, C_{+}>$ or another edit pair $<C'_{-}, C'_{+}>$), we generate one tree edit at a time step...".
So, since in the gold dataset  $<C_{-}, C_{+}>$=$<C'_{-}, C'_{+}>$ , the authors model the "actions leading to $C_{+}$" given "an encoding of $C_{+}$"?
I.e., are the output labels (indirectly) encoded in the input?

This is fine if we consider the gold datasets as intermediary/intrinsic objectives, and consider Fixers one-shot as the "important", downstream task.
If so, Graph2Tree (Yin et al., 2019) performs best on the Fixers one-shot dataset (which is the "important" dataset).
If not, and the gold datasets are meaningful on their own, then why there is no comparison to the seq2seq editor+decoder that Yin et al., 2019 found to perform best on the gold datasets?

2. The paper states that the graph edit encoder of Yin et al. (2019) does "not explicitly express the differences between the input and the output trees". As it looks in Yin 2019 (Section 3.2), it seems that they represent the difference between the trees pretty explicitly, using edges such as "Removed", "Added" and "Replaced" between the old and the new tree. So, what is the main novelty compared to Yin et al. (2019) in this area?

3. The main novelty in this paper compared to previous work, in my opinion, is the imitation learning training (Section 4). I wish the authors elaborated more on this, give examples, show how the parameter values affect the performance, etc.

4. How long are the edit sequences, average, in the Fixers dataset? (in comparison to Hoppity's 3 edits per sequence)

### Improving the Paper ###
The paper could be improved in the following ways:
1. A conceptual discussion of the differences from previous work (Tarlow, Dinella, Brody).

2. Is it possible to perform an empirical comparison with Hoppity?

3. Other strong baselines would be Transformer+copy and bidirectional LSTM seq2seq+attention+copy mechanism, that copy individual tokens rather than spans as in CopySpan (Panthaplackel et al., 2020a).
Even Section 5.2 of the paper says that the CopySpan baseline, which can copy large spans, memorizes too much.
Another straightforward baseline is a sequential **tagger**, that tags each individual token in $C_{-}$ with tags like KEEP/SWAP/DELETE, as in Malmi et al. (2019), and as Brody et al. (2020) adapted to code as their baseline.

4. Adding more interesting examples in which the model succeeds (and fails). The paper claims "applicability to much longer edit sequences" (Section 1), but the example in Figure 1 converts `list.ElementAt(i+1)` to `list[i+1]`   (which might be syntactically-long, but not that "difficult") and the example in Table 2 only converts a `VAR2.ToString()` to `VAR2`  (which is also not very "difficult" or "interesting", because a simple sequence-tagger that is trained to delete tokens can produce such examples).  Ideally, the appendix could contain many additional examples like Table 2 (but also with longer edit sequences).

### Minor questions and comments ###
5. The paper's title contains the phrase "*Incremental* Tree Transformations".
What is "incremental" about it? Doesn't the model mainly model a single "action" or "transformation" at a time? Doesn't the "incrementality" come from an LSTM "decoder" that sequentially predicts these actions?

6. In Section 3.2 - Why is $a_{Stop}$ computed as $W_{Stop}emb(Stop)+b_{Stop}$, instead of just learning a single embedding vector, if all its components are trainable and used only in $a_{Stop}$?

7. Typo in Section 4, Algorithm 1 and Algorithm 2:  "Samping" -> "Sampling"

---

> ### Author Response · Authors · 2020-11-18
> **Thanks for your comments!**
>
> We thank Reviewer 1 for recognizing our work as presenting “strong empirical results and a combination of interesting ideas”! We also feel excited that Reviewer 1 finds our imitation learning idea interesting and novel!
>
> **The overall novelty of our work**
>
> We would like to emphasize that our work has made technical contributions from several angles. Please refer to **“To All Reviewers”** for a detailed discussion, including a conceptual comparison with previous work.
>
> **The novelty of our proposed edit encoder compared with Yin et al. (2019)**
>
> We clarify that our proposed edit encoder is a brand new one and has a _completely different model architecture_ compared with existing tree edit encoders, including the graph edit encoder proposed by Yin et al. (2019). Specifically, as the reviewer mentioned, Yin et al. (2019) calculated the edit representation by connecting the old and the new tree via “Removed/Added/Replaced”-labeled edges and then running a graph neural network over the two trees jointly. From a “model learning” perspective, we argue this way of expressing tree differences is still implicit, as the edit encoder has simply treated the labeled edges as “yet another feature” in addition to the nodes/edges separately provided by the two trees. Our proposed edit encoder instead has completely shifted the modeling focus to _the targeted edit actions themselves_; it directly (1) learns a vector representation for each edit action in the targeted tree edit sequence and (2) calculates the edit representation as an association of the learned targeted edit action representations.
>
> The difference could also be told from our empirical results. Comparing “Graph2Tree + TreeDiff Edit” in our Tab. 1 with “Graph2Tree + Graph Edit Encoder” in Tab. 4 of Yin et al. (2019), it is obvious that our proposed TreeDiff Edit Encoder gives much better editing accuracy (68.09% vs. 48.05% on GHE-gold; Yin et al. did not provide results of Graph Edit Encoder on Fixers settings for comparison, although we believe our edit encoder will still surpass it).
>
> We will revise our writing to make the comparison and the description more precise.
>
>
> **Evaluation: which datasets should we really care about and which are the main results**
>
> We clarify that **all datasets have their values and one should interpret a model from its empirical results on all settings**. _First_, admittedly Fixers-one shot is a more realistic benchmark; however, we’d like to note that GHE/Fixers-gold has its own merits, as it is a more _controllable_ setting towards answering the question of “how well an editor could performance given a _gold_ edit representation f_\delta(C-, C+)” (Yin et al., 2019). Evaluating on GHE-gold also facilitates direct comparison with prior work in this line.
>
> _Second_, we note that the models are simply _tested_ on Fixers-one shot; all models need to be first _pretrained_ on GHE-gold. Arguably a more reasonable experiment should have both _trained_ and _tested_ a model on the Fixers-one shot setting. However, we are unable to do this because the dataset is too small (with only 2,878 examples).
>
> _Finally_, Fixers only covers a restricted set of 16 code edit categories. Although those edits are the most common ones (since they are supported by existing commercial fixers), they certainly could not cover all the real-world diverse patterns of code edits on the GitHub commit stream. In this sense, the GHE benchmark is still important as a dataset with _much higher coverage_ of realistic edit patterns. We believe collecting a larger Fixers dataset for end-to-end evaluation of neural editors would be an important future avenue.
>
> Based on the above discussion, we clarify the **empirical comparison of Graph2Edit vs. Graph2Tree**. We note three crucial observations: (1) Graph2Edit _substantially outperforms_ Graph2Tree on Fixers-one shot when they are paired with Seq Edit Encoder. This implies that our proposed Graph2Edit editor can drive the edit encoder to learn more generalizable edit semantics. (2) When the two editors are paired with TreeDiff Edit Encoder, their performance on Fixers-one shot is basically _comparable_. It is true that Graph2Tree is slightly better. However, as we discussed, the models are actually trained on the GHE-gold training set. Under this restricted setting, Graph2Edit may have overfitted to the specific edit representation when TreeDiff Edit Encoder directly encodes the target edit sequence in training. We hypothesize that when all models are also trained on the realistic one-shot setting, our Graph2Edit model will outperform Graph2Tree. (3) Note that on the “gold” edit representation settings (especially Fixers-gold), Graph2Edit+TreeDiff clearly outperforms others.
>
> ... (see our follow-up comment)

---

> > ### Author Response · Authors · 2020-11-18
> > **Thanks for your comments! (Cont'd)**
> >
> > ...
> >
> > In summary, we agree with the reviewer that the current experimental setting can be improved to make the comparison more clear. However, when analyzing and comparing various models _on all experimental settings_, it is reasonable to claim the superiority of our proposed editor and edit encoder.
> >
> > We emphasize that the topic we focus on, i.e., editing tree-structured data triggered by an edit specification (in the form of code edit pairs), is very new, and research about this topic is just getting started. At the time we started our study, the datasets and experimental settings by Yin et al. (2019) are among the best that we believe would help us understand the problem setting and develop innovative models and learning algorithms. We therefore have mostly followed their settings in our paper. In the future, we would like to further explore more realistic experimental settings and create new datasets to complement research in this direction.
> >
> >
> > **Other questions/comments**
> >
> > Above, we have tried to respond to the reviewer’s most important questions and comments on our work. For the remaining ones:
> > 1. “I wish the authors elaborated more on this (imitation learning), give examples, show how the parameter values affect the performance, etc.”: We will add more details and examples in either our main content or the appendix. In short, when applying the DAgger algorithm to our tree-structured data editing problem, we found out that the model performance very depends on its beta parameter, which controls how often that the model samples “states” from the expert. Our model performs better when this sampling is more frequent (e.g., beta=0.5).
> > 2. “How long are the edit sequences, average, in the Fixers dataset?”: It is 7.089, when the CopySubTree operation is allowed. Note that without the subtree copying (i.e., if the editor has to add individual nodes in a subtree one by one), the average edit length on Fixers will be even longer.
> > 3. “Other strong baselines”: We thank the reviewer for the suggestion. We did not include the baseline of seq2seq that copies single tokens because this model has been shown much weaker than the CopySpan baseline in Panthaplackel et al. (2020), under the same datasets. We will consider adding a sequential tagger model if the rebuttal time allows. We hypothesize that our model will perform stronger than the tagger model because the latter does not utilize any grammar constraint. As we investigated CopySpan in our experiments, we found that such sequential models are prone to produce ungrammatical programs.
> > 4. “Adding more interesting examples in which the model succeeds (and fails). The paper claims "applicability to much longer edit sequences"...”: Please refer to our discussion in **“To All Reviewers”** for a clarification of our intention in mentioning the “longer edit sequences”. However, we agree with the reviewer to provide more examples in the appendix for interested readers.
> > 5. About “incremental tree transformations” in our title: It describes that our model can _iteratively_ produce tree edits and apply the edits to transform the tree. It is true that the “tree edit decoder” (Sec 3.1.3) decodes one tree edit at a time, but our complete tree editor (including both the tree and tree history encoder as well as the tree edit decoder) has been able to transform trees incrementally.
> > 6. About the calculation of a_{Stop}: This design choice is rather optional and we hypothesize that it will not change the model performance a lot.

---

> > > ### Comment · AnonReviewer1 · 2020-11-18
> > > **AnonReviewer1**
> > >
> > > Thank you for your response.
> > >
> > > I really like this paper, I understand the technical contributions and appreciate the engineering efforts, and **I am increasing my score to 7**, assuming that the previous points (discussion, comparison, contributions, more examples, etc.) will be included in the revised version within this discussion period.
> > >
> > > However, I am still confused about the evaluation.
> > > What is a convincing benchmark that shows that this model performs better than previous work?
> > >
> > >
> > > > First, admittedly Fixers-one shot is a more realistic benchmark...
> > > > GHE/Fixers-gold has its own merits, as it is a more controllable setting towards answering the question of “how well an editor could performance given a gold edit representation..."
> > >
> > > Is the question "how well an editor could perform given a gold edit" relevant?
> > > Isn't this question equivalent to the question "how well can a machine learning model *predict the label given the label*?"
> > >
> > > > Second, ... the models are simply tested on Fixers-one shot.. a more reasonable experiment should ...
> > >
> > > I understand these practical considerations, but currently, the paper did not convince me that Graph2Edit is superior over the baselines in the training domain ("gold"), nor in the test domain (Fixers).
> > >
> > > In the GHE-gold dataset ("the training domain") - the best approach here achieves 70.44% accuracy.
> > > Yin et al. 2019 showed (in Table 4) the best accuracy of 59.63% for the seq2seq model. Are these numbers comparable? If so, why can't the authors include these baselines?
> > >
> > > In the Fixers-one-short dataset ("the test domain") - the best approach (Yin 2019) achieves 42.0 / 47.48.
> > > However, Table 5 of Yin 2019 shows that "Graph2Tree - Seq Edit Encoder" achieves 49.21 which is higher but not included nor discussed.
> > >
> > >
> > > >Finally, ... the GHE benchmark is still important as a dataset with much higher coverage of realistic edit patterns
> > >
> > > I agree, this is an interesting and important benchmark, but it is unrealistic in the sense that the model (indirectly) sees the label at test time. Aren't the results on this dataset equivalent to **comparing training accuracy**? (rather than test accuracy?)
> > >
> > > Is it possible to compare different models on the GHE benchmark while *hiding* the label or without edit encoder?
> > >
> > > > Seq2seq baselines
> > >
> > > In Table 4 of Yin 2019, the "Seq2seq - Seq Edit Encoder" achieves 59.63 Acc@1 at GHE.
> > > Is this number comparable to the numbers in Table 1? If so, can it be included in Table 1?
> > >
> > > > the baseline of seq2seq that copies single tokens because this model has been shown much weaker than the CopySpan baseline in Panthaplackel et al. (2020), under the same datasets
> > >
> > > Can it be included in this paper's tables, with a reference to where the results were taken from?
> > >
> > > > About “incremental tree transformations” in our title: It describes that our model can iteratively produce tree edits and apply the edits to transform the tree
> > >
> > > This is minor and did not affect my score, but I think that the word "incremental" in the title is redundant, and emphasizes the incrementality rather than the actual novelty in this paper.

---

> > > > ### Author Response · Authors · 2020-11-20
> > > > **Thank you again for your comments and for increasing your score!**
> > > >
> > > > Thank you again for recognizing our contributions and for increasing your score! We will definitely include all the points that we have discussed in the revised version of our paper (in either the main content or the appendix, depending on the space).
> > > >
> > > > We are also happy to address your other concerns regarding the evaluation.
> > > >
> > > > **Further discussion about the GHE benchmark**
> > > >
> > > > Q: Is the question "how well an editor could perform given a gold edit" relevant? Isn't this question equivalent to the question "how well can a machine learning model predict the label given the label?"
> > > >
> > > > A: This is certainly a reasonable point, but we would argue that the gold setting is still interesting to evaluate, especially given the fact that all of our edit representations consist of a single continuous vector (regardless of the length or nature of the structural edit sequence). Because of this the edit representation is “bottlenecked” and it is thus non-trivial to remember and utilize the original edit sequence, even if it is given as the input. Because of this, models that are able to reproduce the desired output effectively have a demonstrably better inductive bias that allows them to do so efficiently. This was the original motivation expressed by Yin et al. (2019) when they devised this setup, and we also believe that it is relevant in the present paper as well.
> > > >
> > > > Q: Aren't the results on this dataset equivalent to comparing training accuracy? (rather than test accuracy?)
> > > >
> > > > A: Not really. The test-time examples are different from the training time ones, and especially given the diversity of GHE and the above-mentioned continuous vector bottleneck, creating a model that generalizes to the test-data, even in the gold setting, is non-trivial. This is evidenced by the fact that previous work is not able to do so as effectively as our methods.
> > > >
> > > >
> > > > Q: Is it possible to compare different models on the GHE benchmark while hiding the label or without edit encoder?
> > > >
> > > > A: It is possible to do so, but GHE is a setting where the edits are triggered by (and also semantically constrained by) a certain intent (such as refactoring coding style or adding functionality). Requiring the editor to perform edits without specifying the type of edit to be performed would result in an arbitrary number of valid edits, which would add substantial noise to the evaluation process.
> > > >
> > > > **Clarification of the comparison with existing approaches**
> > > >
> > > > Sorry for the confusion in our comparison with existing approaches!
> > > >
> > > > We did not include all existing approaches because some of them have been reported to be weaker than the others we did test under the same setting. For example, “Seq2Seq + Seq Edit” performs worse than “CopySpan + Seq Edit” on both GHE-gold and Fixers-one shot (micro) in Panthaplackel et al. (2020); Yin et al. (2019) claimed that “Graph2Tree + Seq Edit” is better than both “Seq2Seq + Seq Edit” and “Graph2Tree + Graph Edit”. It is possible to still include these approaches in our table, however, as we have adjusted the evaluation settings slightly (e.g., adding Fixers-gold and changing the evaluation procedure of Fixers-one shot -- we will explain this shortly), results are largely missing for existing approaches. Therefore, we only re-tested and compared with the existing state-of-the-art approaches (CopySpan and Graph2Tree) in our experiments.
> > > >
> > > > We note that our calculation of the Fixers-one shot accuracy follows Panthaplackel et al. (2020), which is different from the calculation procedure used by Yin et al. (2019). Specifically, for each fixer category, Yin et al. (2019) report the best accuracy score that a model can achieve among 10 “seed” representations, while we report an average accuracy over 100 seed representations (see Sec 5.1 for details; we will also clarify this in our revised paper). We believe enlarging the number of “seed” representations and reporting the average performance of a model can give a more reasonable and reliable comparison, let alone that, as Panthaplackel et al. (2020) pointed out, the calculation procedure of Yin et al. (2019) involves random sampling, which may not be reproducible. Given this change, numbers reported in Table 5 of Yin et al. (2019) are not comparable with our Fixers-one shot evaluation.
> > > >
> > > > In the following table, we retrieve back some of the existing approaches. Notation: * indicates that numbers are copied from the original paper; -- indicates missing evaluation that is not provided in the original paper; other numbers without indication are reported by us. For baseline models that are re-tested by us, we use implementations kindly provided by their authors and make sure that our reproduced version has comparable performance as what was reported in their paper. In our revised draft, we will give a clarification of the comparison and consider adding all these approaches to Table 1.
> > > >
> > > > ... (see the follow-up comment)

---

> > > > > ### Author Response · Authors · 2020-11-20
> > > > > **Thank you again for your comments and for increasing your score! (Cont'd)**
> > > > >
> > > > > | Model  | GHE-gold  | Fixers-gold   | Fixers-one shot (macro) | Fixers-one shot (micro) |
> > > > > |---|---|----|---|---|
> > > > > |  Seq2Seq + Seq Edit (Yin et al., 2019) |  59.63* | -- | --  | --  |
> > > > > |  Seq2Seq + Seq Edit (Reimplemented w/ tricks by Panthaplackel et al., 2020) |  64.4* | -- | --  | 18.8*  |
> > > > > |  CopySpan + Seq Edit (Panthaplackel et al., 2020) |  67.4* |  87.07  | 20.64  | 24.2*  |
> > > > > |  Graph2Tree + Seq Edit (Yin et al., 2019) |  57.49* |  81.34  | 33.94  | 40.55  |
> > > > > | Graph2Edit + Seq Edit (Ours)  |  54.69 |  70.50  | 41.82  | 45.58  |
> > > > > |  Graph2Tree + Graph Edit (Yin et al., 2019) | 48.05*  |  --  | --  | --  |
> > > > > |  Graph2Tree + TreeDiff Edit (w/ our edit encoder) |  68.09 |  82.94  | 42.00  | 47.48  |
> > > > > | Graph2Edit + TreeDiff Edit (Ours) | 70.44 | 92.32 | 41.21 | 44.75|

---

> > > > > > ### Comment · AnonReviewer1 · 2020-11-20
> > > > > > **Thank you for your response**
> > > > > >
> > > > > > Thank you for your response,
> > > > > > I have one additional question:
> > > > > >
> > > > > > >we would argue that the gold setting is still interesting to evaluate
> > > > > >
> > > > > > I agree that it is still interesting to evaluate.
> > > > > > Do you see the gold setting as a task that can be realized in realistic settings, or is this only a "learning benchmark"?
> > > > > >
> > > > > > If this is mostly a learning benchmark - I understand; but in realistic settings (e.g., as an IDE / GitHub tool) - the model will not have the edit representation as input (because that would be the desired target).

---

> > > > > > > ### Author Response · Authors · 2020-11-20
> > > > > > > **Thanks for your prompt reply!**
> > > > > > >
> > > > > > > Thank you for your prompt reply! We agree that the gold setting is more like a learning benchmark. In realistic applications, the user could only provide similar edits on other inputs (close to the Fixers-one shot setting).
> > > > > > >
> > > > > > > We are happy to address more questions if there are any.

---

> > > > > > > > ### Comment · AnonReviewer1 · 2020-11-20
> > > > > > > > **Increasing my score**
> > > > > > > >
> > > > > > > > Thank you for your response.
> > > > > > > >
> > > > > > > > **I am increasing my score to 8**.

---

> > > > > > > > > ### Author Response · Authors · 2020-11-20
> > > > > > > > > **Thank you for increasing the score!**
> > > > > > > > >
> > > > > > > > > Thank you for increasing your score! This is a great encouragement for us to pursue the study. We will have our revised paper ready soon.

---

### Official Review · AnonReviewer3 · 2020-10-26
**Impressive general model with some problems in the experiments**

**Rating:** 7
**Confidence:** 4

**Review:**

## Summary

The paper proposes a general model for incremental editing of tree-structured
data such as abstract syntax trees. The editing operations include adding a
node, deleting a subtree, or copying a subtree. They also propose a novel edit
encoder to learn to represent edits, and an imitation learning method to make
the model more robust.

## Pros

- The work has several interesting and valuable contributions:
  + Compared to previous work, the model is much more general: it supports
    general tree edits, is language agnostic, and can handle much longer edit
    sequences.
  + The novel edit encoder which directly encodes the edit actions is more
    intuitively correct and also performs better than previous approaches.
  + Imitation learning to make the model more robust is a natural idea that
    suits incremental edits very well.
- The source code will be released which -- given the general nature of the
  model -- could enable further interesting research.

## Concerns

- The explanation for the results in Table 1 could be improved upon.
  + I don't think the results support the claim that Seq Edit Encoder memorizes
    specific patterns with the baselines as the micro average of Graph2Tree for
    Fixers-one shot is very close to Graph2Edit.
  + Even if we accept the claim, that would not explain the CopySpan >
    Graph2Tree > Graph2Edit results on GHE-gold and Fixers-gold. The authors
    state that Seq Edit memorizes specific patterns, TreeDiff Edit learns more
    generalizable information, and Graph2Edit makes Seq Edit learn more
    generalizable information. But they state that in order to solve GHE-gold,
    we need specific patterns instead of generalization. So why does increasing
    generalization by switching Seq Edit to TreeDiff Edit improve performance of
    Graph2Edit on GHE-gold and Fixers-gold (which doesn't need generalization),
    and decrease performance on Fixers-one shot (which needs generalization)?
- Some of the contributions are not demonstrated clearly:
  + The method's applicability to much longer edit sequences is not clearly
    demonstrated, although the length of the edit sequences is mentioned briefly
    in the Appendix.
  + The imitation learning method is demonstrated only on "Graph2Edit with Seq
    Edit Encoder", which is the less interesting case compared to "Graph2Edit
    with TreeDiff Edit Encoder". From the previous experiment it's clear that
    the TreeDiff encoder version is the practically relevant one and the one the
    paper's about. As the Seq Edit encoder makes more mistakes and so it's
    easier to improve, we don't know whether imitation learning helps in the
    relevant case of the TreeDiff encoder. In my opinion this makes imitation
    learning more of a digression and a less of an organic part of the paper.
- The writing could be more precise at times. For example, the authors state
  that they are adding subtrees as an edit operation, but as far as I
  understand, they are adding individual nodes.

## Reasons for ranking

I believe that the model is an important step in learning to represent edits.
However there are some problems with the experiments: some of the claims are not
adequately supported and the explanations could be improved upon.

## Minor comments

- I found Figure 1 confusing at first, because there is essentially no caption
  and the description of the figure comes much later in parts. It would be good
  to either have a more substantial caption or to move the figure closer to the
  explanations.
- Table 2 precedes Table 1, which is confusing
- It should be "general", not "generic", like "general model for incremental editing"
- Page 3: Delete operators take a tree node ... and remove
- The algorithms in the Appendix should be DaggerSampling and PostRefineSampling (missing "l").

---

> ### Author Response · Authors · 2020-11-17
> **Thanks for your comments!**
>
> We thank Reviewer 3 for recognizing our contributions as “interesting and valuable”!
>
> **Explanations for Table 1**
> 1. Clarification of “Seq Edit Encoder memorizes specific patterns (about C+) with the baselines”: We make this conjecture when observing the two baselines, CopySpan and Graph2Tree, show superior performance on GHE/Fixers-gold but worse performance on Fixers-one shot. Intuitively, this is because the target output for the two models, the ground-truth edited code C+, has been exposed to the Seq Edit Encoder.
>
>  However, due to their different architectures, the two models still show different behavior. For Graph2Tree, since it generates C+ _in the form of an AST tree_, it suffers less from the “memorization” or overfitting of the Seq Edit Encoder; this explains its better accuracy on Fixers-one shot. In contrast, since CopySpan generates C+ _in the form of a token sequence, which is exactly the same as how C+ is encoded by the Seq Edit Encoder_, it suffers the most from the “memorization” problem and cannot generalize well to the Fixers-one shot setting.
> 2. Clarification of the performance of “Graph2Edit + TreeDiff Edit Encoder”: We first clarify that the TreeDiff Edit Encoder has the advantages of being both _expressive_ and _generalizable_. The “expressive” explains the improvement of Graph2Tree and Graph2Edit when they are paired with TreeDiff Edit. Second, as we discussed in the end of Sec 5.2, the decreased performance of Graph2Edit on Fixers-one shot is likely because it has overfitted to the specific edit representations during training. Note that all models (including Graph2Edit) are trained on the GHE-gold training set and are only tested on Fixers-one shot. This overfitting is caused by a similar “memorization” issue that CopySpan has, when the target output of the model (ground-truth tree edits for Graph2Edit and ground-truth code tokens for CopySpan) has been exposed to the edit encoder (TreeDiff Edit for Graph2Edit and Seq Edit for CopySpan) in exactly the same format.
>
> **Response regarding some of the contributions not being demonstrated clearly**
> 1. About “longer edit sequences”: please see **“To All Reviewers”** for clarification, as well as other discussions about the overall contribution/novelty of our work.
> 2. About the imitation learning experimental setting: We agree with the reviewer that experimenting with "Graph2Edit with TreeDiff Edit Encoder" would definitely be more interesting and more convincing in supporting our claim. We apologize for not being able to explain this in our initial submission.
>
>  As suggested, we experimented with the setting using “Graph2Edit with Treediff Edit Encoder,” and found that, interestingly, it seems that the model does sufficiently well on the GHE dataset already without imitation learning that further gains through imitation learning were hard to obtain. (In fact, 80% of the remaining errors were due to issues such as unknown tokens, which cannot be fixed with better training algorithms, as they are outside of the search space of our current model.)
>
>  Note that we do _not_ believe that this is necessarily a result discounting the utility of imitation learning in general, but rather just a result of the GHE-gold setting being relatively easy (compared to, for example Fixers-one shot). However, we do not immediately have training data to use for imitation learning in harder settings (Fixers is test data only), so as a proxy for testing on a harder dataset we believe the experiments with the weaker “Graph2Edit with Seq Edit Encoder” model provide a good proxy demonstrating imitation learning’s potential. Experimenting with imitation learning on harder datasets is definitely very high on our list of things to do -- we were not able to do it in the short time span for author response, but if you think it would contribute significantly to the paper we can try to do it for the final version.
>
> **Regarding preciseness of writing**
>
> Thank you for your advice! We will revise Sec 3.1.1 to describe the tree edit actions more precisely. To clarify, the CopySubTree operator copies the whole subtree from g_1 (the initial tree) to g_t (the current tree at time step t) in one single step, rather than “adding individual nodes” in multiple steps. An example is shown in Fig 1(b) at t=4, where the whole subtree “Expr → i + 1” is copied from the input tree g_1 in Fig 1(a). In our model architecture, the tree edit decoder decides which subtree to copy from g_1, by using the learned representation of the root node of this subtree (i.e., the node representation of “Expr” in the running example) as a feature. Due to space constraints, we have only briefly mentioned this in the last paragraph of Sec 3.1.3, but more details can be found in Appendix A.2.
>
> We also thank you for the minor comments! We will address them in the revised draft.

---

> > ### Comment · AnonReviewer3 · 2020-11-22
> > **Thank you for your response**
> >
> > Thank you for your response! I also read the other discussions, together they answered my questions.
> >
> > I increased my score to 7.
> >
> > For imitation learning, it seems that indeed it could not improve "Graph2Edit with Treediff Edit Encoder". I accept the reasons in your response. Please include this result and the reasons in the paper (or the reasons in the Appendix if there's not enough space in the paper).

---

> > > ### Author Response · Authors · 2020-11-22
> > > **Thank you for increasing your score!**
> > >
> > > Thank you for increasing your score! We will definitely include our result and discussion for imitation learning in the revised paper. We will also be happy to address more questions you have.

---

### Official Review · AnonReviewer2 · 2020-10-28

**Rating:** 7
**Confidence:** 4

**Review:**

The paper presents a neural autoregressive model that learns to incrementally or iteratively perform edit actions on structured data. The authors focus specifically on abstract syntax tree representation of programs (e.g., C#). The model has two main parts:

(1) Neural editor models p(a|) that iteratively performs tree edit actions such as sub-tree deleting or adding;
(2) Edit encoder learns edit representations f_Delta by encoding the sequence of ground truth tree edit actions.

The authors also propose an imitation learning algorithm to train the editor and evaluate the model on source code edit datasets.

The idea of incremental tree transformation pretty much follows that of Hoppity by Dinella et al, ICLR 2020 for bug fixing of Javascript programs. Unlike Hoppity where each transformation step is done on a single node, this work extends to support sub-tree operations. However, adding a subtree is nothing but performing a sequence of single-node actions. The difference is that to ensure the syntactic validity of the tree at any point, the authors use a grammar specified a priori. This mechanism was proposed in semantic parsing by Yin et al., ACL 2017. Compared to Hoppity, this model has another SubTreeCopy operation, but it is a somewhat straightforward extension of the copy mechanism from Yin et al.

Although the whole idea is not new, I think the paper presents a valued extension to existing work. The running example is intuitive, making the paper easy to follow. However, there are a number of parts that are unclear and need more clarity. I hope the authors address these during the rebuttal phase.

1. The authors should be clear (e.g., before Equation 1 and in Section 3) about where the sequence actions {a_t} comes from and how sub-tree actions are represented (e.g., decomposed into single-node actions). Later in this experiment, the authors mention dynamic programming to compute the shortest tree edit sequence, but I feel that substantial discussion or an algorithmic description is needed.

2. It is not clear to me whether f_Delta is learned jointly or separately from the parameters of p(a|). In Figure 1 and Equation 1, f_Delta does not depend on t and seems to be fixed.

3. In the last paragraph of page 3, this sentence “the operator selects a dummy node (e.g. node Dummy) and replaces it with the added node” is not clear. Is there always at least one dummy node at any time? What is “the added node”?

4. Another question about Add operation: if an Add operation is given and there is currently no dummy node, would it make more sense to specify the added location with respect to a parent node and its children?

5. How does the model know when to stop adding a right sibling for constructor fields with *sequential* cardinality? Similarly, for an *optional* cardinality field, what does the model do on the attached dummy node if the field is indeed optional?

6. It may be more natural to predict the node location n_t before the operation op_t. Equation 2 does the other way. Does this change the model in any way?

7. To what extent this framework is language-agnostic? Despite the ASDL, even for the same language, different parsers can have different grammar specifications, so how easy is it to apply this framework for other languages? On the same note, since Hoppity is directly related, I am curious how this work compares to it on a same bug fixing task.

Finally about the experiments, the authors compare to Graph2Tree and CopySpan. But it does not look like the source code for the baselines are available. Releasing the source code with that of the baselines would be helpful.


Minor typos

“Arbitary”: Section 3.1.1 paragraph 2.

========== After discussion =============

I have increased my score from 6 to 7.

---

> ### Author Response · Authors · 2020-11-17
> **Thanks for your comments!**
>
> We thank Reviewer 2 for recognizing the value of our work and giving the detailed comments!
>
> **Answers to reviewer’s questions**
> 1. Thanks for your advice! We will revise our draft to make clear the actions’ definitions and where they come from. To clarify, the CopySubTree operator copies a _complete_ subtree from the initial input tree (g_1) to the current tree (g_t) in one single step, rather than decomposing it into multiple steps of adding tree nodes. An example is shown in Fig 1(b) at t=4, where the whole subtree “Expr → i + 1” is copied from the input tree g_1 in Fig 1(a).
>
>  The dynamic programming algorithm mentioned in Sec 3.3 is used only for calculating gold-standard edit sequences in our training data. We will add a pseudo-code of the algorithm in the revised version.
> 2. As we mentioned at the beginning of Sec 3.3, the edit encoder (which produces the edit representation f_delta) is jointly trained with the editor. It is correct that f_delta does not depend on the time step t, but it is a non-fixed, learnable vector representation.
> 3. A dummy node denotes a vacant position that is syntactically valid to accept a tree node (see node “Dummy” in Fig 1(b) at t=1 for example). It is automatically added following the underlying grammar to ensure no missing child for each tree node. From another perspective, this also means that it is possible to _not_ have dummy nodes, e.g., when all tree nodes’ syntactically valid child nodes have been fulfilled. **Note that this will never happen if any tree node has a _sequential_ field, because we always append one dummy node to a sequential field and a sequential field can have a new child inserted to every possible child position in any time.**
>
>  For our editor, **in the case of single/optional fields,** an Add operator has to be applied to a dummy node. This is intuitive since the Add operation means to add a tree node to a certain syntactically valid position in the current tree, and all such positions **for single/optional fields** have been held by dummy nodes.
>
>  The “added node” in the sentence refers to a _non-terminal_ node such as node “ElementAccess” in Fig 1(b) at t=2. Predicting a non-terminal to-be-added node is equivalent to selecting a production rule (e.g., AssignStmt → ElementAccess) for the node’s parent (e.g., AssignStmt). Note that the Add operator can also be used to populate an empty _terminal_ node with a literal value (e.g., token “list” at t=3). In this case, deciding the terminal node to be added is equivalent to predicting a token in place.
> 4. Please refer to our answer “3.” for an explanation about the dummy node mechanism. To this question: since all available positions that are syntactically allowed to accept tree nodes **for single/optional fields** have been replaced by dummy nodes in our mechanism, having no dummy node means having no valid position for adding nodes **to single/optional fields** in the current tree. In this case, an Add operator is illegitimate **for such fields** and will be eliminated from the operator candidates.
>
>  **For nodes with sequential fields, there will always be a dummy node appended to it (e.g., [A, B, Dummy]). To add a new node (e.g., node C) to a certain position of a sequential field, our model first selects the “right sibling node of the target position”, and then inserts the new node to its left. For example, adding C before A is done by selecting A and inserting C before it; adding C as the actual rightmost child is done by selecting Dummy and inserting C before it.**
>
> 5. Due to space constraints, we discuss details about ASDL implementation in Appendix A.1. For fields with sequential cardinality, there is always one extra dummy node attached as its rightmost child. For example, the child list [A, B] is extended to [A, B, Dummy]. Therefore, our editor always has the option to add a right sibling to the child list of a sequential-cardinality field. This is implemented by selecting Dummy and replacing it with the to-be-added tree node, which will extend the child list into [A, B, C, Dummy] (with C being the added node). **Note that this is the same as first selecting Dummy and then inserting node C to its left, which we described in the edited answer “4”.** “When to stop adding a right sibling” is left as a learning problem to the neural editor itself.
>
>  For optional-cardinality fields, a dummy node is attached when their child position is not taken. To add a child node to the field, our editor similarly replaces the dummy node with the node to be added. When the field has already gotten a non-Dummy child, no more dummy child node is attached to ensure its grammatical correctness.
> 6. We decide the operator prior to the node location simply because for the operator Stop, selecting whatever node location is meaningless. Following our current design, in inference time, once the editor chooses the Stop operator, it has no need to pick the node location, which is more natural than the other way.
>
> ...

---

> > ### Author Response · Authors · 2020-11-17
> > **Thanks for your comments! (Cont'd)**
> >
> > ...
> >
> > 7. Please refer to **“To All Reviewers”** for a detailed discussion about the novelty of our work, including its language agnosticism advantage and a comparison with Hoppity. In short, building our editor at the top of ASDL has made it being language-agnostic in its nature.
> >
> >  Regarding “how easy is it to apply this framework for other languages”: We note that our framework has decoupled the language grammar specifications from the model architecture -- the former is handled by ASDL and the latter corresponds to our neural network implementation (which does not involve language-specific designs). As demonstrated by Rabinovich et al. (2017) and Yin & Neubig (2018), ASDL can be flexibly applied to handle grammar specifications from different languages. Therefore, we conclude that it should also be easy to apply our framework for other languages. Although, inevitably, it might still require some engineering effort, such effort is relatively small, and only has to be done once when applying our framework to various datasets in the same language.
> >
> > **Source code release**
> >
> > In experiments, all baselines are implemented based on the source code kindly provided by their original authors. Upon the acceptance of our submission, we will release the implementation of our model.

---

> > > ### Comment · AnonReviewer2 · 2020-11-20
> > > **Further clarification needed**
> > >
> > > Thanks the authors for answering my questions. I'd like to discuss some of the points further.
> > >
> > > About 3 and 4, I've had no confusion about the dummy node mechanism, and it is clear in the illustrated example. What's unclear to me is in more complicated examples, such as body of a method declaration or a block, etc. In these cases, the corresponding AST of the original method is syntactically valid, but one can always add one or more statements. The new statements can be added at different positions in the body, so it is unclear how the dummy node mechanism handles these cases, which I think are very common practical scenarios.
> > >
> > > In the response, the authors mention "since all available positions that are syntactically allowed to accept tree nodes have been replaced by dummy nodes in our mechanism..." If I remember correctly, this is not discussed in the paper. Moreover, it has not yet fully addressed my concerns I just clarified above.
> > >
> > > Regarding 5 and related to 1, the authors say "“When to stop adding a right sibling” is left as a learning problem to the neural editor itself." I can see this only if there are explicit signals in the gold-standard edit sequences to teach the model to learn to stop. But again, the discussion of action sequences is too ambiguous in the current version. Please include some examples of gold-standard sequences.
> > >
> > > My another question: are there any cases where a field is both *optional* and *sequential*. How would the grammar and the model (or the dummy node mechanism) handle such a case?
> > >
> > > Finally, I am willing to change my score, depending how my concerns are addressed in the revised version.

---

> > > > ### Author Response · Authors · 2020-11-21
> > > > **Thank you for your reply!**
> > > >
> > > > Thank you for your reply and your willingness to change your score! We are happy to address your concerns in both the author response and the reviser paper.
> > > >
> > > > **Further clarification of the dummy node mechanism**
> > > >
> > > > We apologize that we did not explain the dummy node mechanism clearly in either the initial paper or the previous author response. We would like to further clarify this mechanism especially for adding new nodes or copying subtrees to sequential fields.
> > > >
> > > > Consider a “block” containing two statements A and B, which are grouped under a **sequential** field “Statements”. Since our mechanism always attaches a dummy node to a sequential field, the field of this block now has a child list of [A, B, Dummy]. Adding a new statement (i.e., a new tree node) to a certain child position of this field is performed by first selecting the _right sibling of the target position_ and then inserting the new node to its _left_. For example, to add a node to the _left_ of A is achieved by selecting node A and inserting the new node before it, and adding a node to the _right_ of B (i.e., as the rightmost _actual_ child) is done by selecting the Dummy node and inserting the new node before Dummy. Copying a subtree to a sequential field follows the same procedure, except that it is a complete subtree rather than a single tree node being inserted to the target position.
> > > >
> > > > For **single** and **optional** fields, adding a new node or copying a subtree to its child position has to be applied to a Dummy node, which indicates the only vacant child position for the fields. This is achieved by replacing the Dummy node with the new node/subtree.
> > > >
> > > > In our initial paper, we have briefly described this mechanism for each kind of field in Appendix A.1. However, in our main content (Sec 3.1.1), for ease of explanation, we have only introduced the case of single/optional fields (that’s why we describe “to add a non-terminal node, the operator selects a dummy node and _replaces_ it with the added node” in Sec 3.1.1). We will revise this paragraph in the new version.
> > > >
> > > > We are sorry for the confusion in our previous response (answer 3, 4 and possibly part of 5). We overlooked the case of sequential fields and thought you were only asking about the single/optional-field cases that we describe in Sec 3.1.1. **We have revised our previous answers in the last comment for clarification and have marked the changes in bold.** Please let us know if it is clear now.
> > > >
> > > > **Clarification of “when to stop”**
> > > >
> > > > Yes, a global Stop action is included as the last action in every gold-standard edit action sequence in our training set, which will be used for training and is generated by our dynamic programming algorithm (Sec 3.3). Therefore, our model can be trained to decide when to stop the editing process.
> > > >
> > > > We also note that the “Stop” action signals the end of editing _the whole tree globally_. We don’t have an explicit Stop signal for editing each sequential/optional field within the tree. However, this has been implicitly indicated by the global Stop action, i.e., as the model learns to stop editing the whole tree globally, it actually also learns to stop editing any certain fields within it.
> > > >
> > > > An example of the gold-standard edit action sequence can be found in Figure 1(b):
> > > > - a_1 = apply the Delete operator at t=1 to remove node MethodCall (and its descendants);
> > > > - a_2 = apply the Add operator at t=2 to add a non-terminal node ElementAccess to the target position indicated by node Dummy;
> > > > - a_3 = apply the Add operator at t=3 to add a terminal node (token) “list” to the target position indicated by node Dummy1;
> > > > - a_4 = apply the CopySubTree operator at t=4 to copy a subtree “Expr → i+1” from g_1 to the target position indicated by node Dummy2;
> > > > - a_5 = apply the Stop operator at t=5 to terminate the editing (which is omitted from the figure).
> > > >
> > > > **Are there any cases where a field is both optional and sequential**
> > > >
> > > > We believe there is not such a field, because the optional cardinality is a special case of the sequential cardinality with maximum length one. More specifically, an optional field can have _zero or one_ child, while a sequential field can have _zero or more_ children (Rabinovich et al., 2017). We did not observe any of such cases for C# in our training set either. Our mechanism can handle both kinds of cardinality, as we explained before.

---

> > > > > ### Comment · AnonReviewer2 · 2020-11-21
> > > > > **Discussion (cont'd)**
> > > > >
> > > > > Thank you for the clarification.
> > > > >
> > > > > In the end, for **"our model first selects the “right sibling node of the target position”, and then inserts the new node to its left..."**, wouldn't the authors need a pointer network to predict the right sibling?
> > > > >
> > > > > **An example of the gold-standard edit action sequence can be found in Figure 1(b):**
> > > > >
> > > > > This example is fine but a little simplistic. Let us take a slightly different version in Figure 1(b) as an example: *"x = list[i+1]"* to *"x = list.ElementAt(i+1)"*. Given the following toy grammar:
> > > > >
> > > > > _ElementAccess(Name obj, Expr index)_,
> > > > >
> > > > > _MethodCall(Expr? obj, Name method, Expr* args)_
> > > > >
> > > > > What is the edit action sequence in this case?

---

> > > > > > ### Author Response · Authors · 2020-11-22
> > > > > > **Response to discussion**
> > > > > >
> > > > > > Thank you for your quick response!
> > > > > >
> > > > > > **Wouldn't the authors need a pointer network to predict the right sibling?**
> > > > > >
> > > > > > Exactly! That is the purpose of the “node selection” module in our “tree edit decoder” (Sec 3.1.3), and this module is a pointer network as you suggested. Let us further clarify all modules in our tree edit decoder and use “adding a new statement (a new tree node) to the sequential field of a block” as an example. For deciding every edit action,
> > > > > > 1. The “operator prediction” module first predicts an operator to perform. In this example, an “Add” operator will be predicted, which triggers the “node selection” module and the “value prediction” module to predict where and what to add, respectively.
> > > > > > 2. Then the “node selection” module decides the target position by selecting an existing tree node from the current tree. In the example (sequential fields), the _“right sibling node of the target position”_ needs to be selected; in the case of adding a tree node to a single/optional field, _the dummy node of the field_ needs to be selected.
> > > > > > 3. Finally, a “value prediction” module predicts the “argument” of the operator. In this example, the argument refers to the new tree node to be added. As we elaborated in answer “3” of our first response, when the new tree node is a non-terminal node, this is equivalent to predicting a production rule for its parent (e.g., assuming a toy grammar of “AssignStmt(Name left, Expr right)”, to derive the field “right” at t=2 in Fig 1(b), the production rule to be predicted is “Expr → ElementAccess(...)”, which ends up with the edge “AssignStmt → ElementAccess” and the new node “ElementAccess”); when it is a terminal node (e.g., string literals), this is equivalent to predicting a token (e.g., token “list” at t=3) from the token vocabulary.
> > > > > >
> > > > > > **A more complicated example of the gold-standard edit action sequence**
> > > > > >
> > > > > > To edit the AST of “x=list[i+1]” to the AST of “x=list.ElementAt(i+1)”:
> > > > > > 1. a_1 = apply a Delete operator to remove node “ElementAccess” (which in effect removes the entire subtree of “list[i+1]” rooted at this node _in a single step_). Note that a dummy node (denoted as “Dummy”) will be automatically added as the right operand of “AssignStmt”.
> > > > > > 2. a_2 = apply an Add operator to add a non-terminal node “MethodCall” to the target position indicated by node Dummy. Since its associated field “right” has a single cardinality, node Dummy will be simply _replaced_ by node MethodCall. Our mechanism will then automatically instantiate child nodes of MethodCall with dummy nodes. Specifically, three dummy nodes are added: “Dummy1” as a child of the field “obj”, “Dummy2” as a child of the field “method”, and “Dummy3” as a child of the field “args”.
> > > > > > 3. a_3 = apply an Add operator to add a terminal node (token) “list” to the target position indicated by node Dummy1. Since its associated “obj” field has an optional cardinality, this terminal token will simply _replace_ node Dummy1. Now the only child position of “obj” is filled. Note that after taking this action, dummy nodes remaining in the tree are Dummy2 and Dummy3.
> > > > > > 4. a_4 = apply an Add operator to add a terminal node (token) “ElementAt” to the target position indicated by node Dummy2. Since its associated “method” field has a single cardinality, this terminal token will simply _replace_ node Dummy2. Now the only child position of “method” is filled. Note that after taking this action, the only dummy node remaining in the tree is Dummy3.
> > > > > > 5. a_5 = apply a CopySubTree operator to copy a subtree “Expr → i + 1” from the input tree to _the target position indicated by Dummy3_. Note that when the associated field “args” has a sequential cardinality, the _“target position indicated by Dummy3”_ is the child position _to the left of Dummy3_. Performing this action will insert the subtree “Expr → i + 1” immediately before Dummy3, leading to a child list of [Expr → i+1, Dummy3] for the field “args”.
> > > > > > 6. a_6 = apply a Stop operator to terminate the editing. Till this moment, all single/optional fields in the tree have been filled and the only dummy node remaining is Dummy3 of the sequential field “args”.

---

> > > > > > > ### Comment · AnonReviewer2 · 2020-11-23
> > > > > > > **Discussion (cont'd)**
> > > > > > >
> > > > > > > Thank you for working out the action sequence of the additional example. My only last point: After step 6, there remains a dummy node, Dummy 3. Would this node be pruned, in a post-processing step or what will the model do with it? In case there are other changes in the input source code after this change “x=list[i+1]” to “x=list.ElementAt(i+1)”, how would the model not pick up from Dummy 3?
> > > > > > >
> > > > > > > Overall, I am satisfied with the authors' response. Please makes changes in the revision, per the discussion. I will update my score to 7 after this.

---

> > > > > > > > ### Author Response · Authors · 2020-11-24
> > > > > > > > **Thank you for your comment and for increasing your score!**
> > > > > > > >
> > > > > > > > Thank you for your comment and for increasing your score!
> > > > > > > >
> > > > > > > > To your question: Yes, we have a post-processing step to remove dummy nodes when the editing process ends. Since we define the dummy node as a special kind of tree node, it is trivial to track and remove them from a tree. In case that other changes are applied to the same tree (after pruning), our mechanism will instantiate a new set of dummy nodes following the same rule that we have explained, such that the new edits are also syntactically valid.
> > > > > > > >
> > > > > > > > We will include all the discussions in the revised paper. We are working on the draft now and will definitely upload it before the end of the rebuttal period.

---

> > > > > > > > > ### Comment · AnonReviewer2 · 2020-11-24
> > > > > > > > > **The answer to my last question is unclear!**
> > > > > > > > >
> > > > > > > > > > **In case that other changes are applied to the same tree (after pruning), our mechanism will instantiate a new set of dummy nodes following the same rule that we have explained, such that the new edits are also syntactically valid.**
> > > > > > > > >
> > > > > > > > > This is not clear. I was not asking about the "after pruning" scenario, because it is simply a different edit example, and you can simply instantiate a new set of dummy nodes. I was talking about continuous changes, such as:
> > > > > > > > >
> > > > > > > > > “x=list[i+1]; sum = sum + x” to “x=list.ElementAt(i+1); sum += x”. How would you get rid of Dummy 3 to start performing edits for the
> > > > > > > > > second statement “sum = sum + x” to “sum += x”?

---

> > > > > > > > > > ### Author Response · Authors · 2020-11-24
> > > > > > > > > > **Further clarification**
> > > > > > > > > >
> > > > > > > > > > Sorry that we misunderstood your question previously.
> > > > > > > > > >
> > > > > > > > > > To clarify, we think this question is more about _whether the model can learn to pick the right edit location_.
> > > > > > > > > >
> > > > > > > > > > In case that further revision such as editing “sum = sum + x” into “sum += x” is needed, there is a chance that a model (especially when it is _not well-trained_) would incorrectly pick Dummy3 as the edit location. However, we expect a _well-trained_ model to be able to correctly identify which operation to perform (via operator predictor) and where/how to perform the operation (via node selector and value predictor). Consequently, it can realize that Dummy3 is irrelevant to the editing of “sum = sum + x” (because Dummy3 does not exist in the AST (sub)tree of this expression) and avoid picking Dummy3 as the edit location.
> > > > > > > > > >
> > > > > > > > > > Please let us know if this is clear. Thank you!

---

### Official Review · AnonReviewer4 · 2020-11-04
**Useful tool perhaps, but claims of paper not demonstrated and novelty unclear.**

**Rating:** 5
**Confidence:** 4

**Review:**

## Summary

This paper presents an approach to learn a model over incremental structural tree edits, that takes as input a partially formed tree and applies a sequence of transformations that edit the tree into a final form.  They focus on abstract syntax trees as used to represent expressions in programming languages, where the grammar is specified using the ASDL formalism.

They demonstrate their approach on two datasets: the GitHubEdits dataset and C#Fixers, showing improved performance relative to some baselines.

## Overview

The motivation for this work is weakly specified.  Analogies are made to the fact that humans seem to edit objects in sequence, but not much more than that is provided.  The authors would do well to specify, or at least hypothesize as to what the benefits of sequences of edits over the alternative.

Writing-wise, it is hard to tell what this paper actually contributes from the abstract or introduction.  Now, I would characterize it as presenting a template in which neural networks can be composed such that graph transformation sequences can be learned from data.

The claimed advantages over existing approaches are:
- Supports general tree edits, whereas previous approaches are somehow more restricted
- Language agnostic due to use of ASDL
- Supports longer sequences of edits in practice

A major problem is that these advantages are only weakly demonstrated, if at all:

- What is a general tree edit?  One answer could be that any graph transformation can be encoded as an action in theri framework, or is it that it can be achieved through a sequence of steps?  The paper focuses on a very particular set of tree edits, so it is very unclear what is being claimed here.
- Longer edit sequences.  No evidence is provided to support this claim.
- Language agnosticism through ASDL is a valid claim, and useful.  It's not clear that generalizing any of the existing work to work with a general grammar would require much work though.

Moreover, the work closest to this work (Dinella et al) is not compared against, neither experimentally, nor are the differences in the approaches detailed.  Why?

Overall while I think a useful software package could be made based on this work (which the authors suggest is forthcoming and which itself could be presented as a publication of a much different kind), the delta over existing work has not been demonstrated enough for me to recommend publication.

## Questions

- In the LSTM encoding of an edit sequence, you have these four transformations to produce a_stop, a_Delete, a_Add, and a_CopySubTree.  What is the advantage of doing these transformations prior to having the LSTM read the result?
- I cannot see where the embedding function $\text{emb}$ is defined.  Is this learned, or fixed?

---

> ### Author Response · Authors · 2020-11-17
> **Thanks for your comments!**
>
> We thank Reviewer 4 for the detailed comments!
>
> **Motivation for this work; the benefits of sequences of edits over the alternative**
>
> We appreciate the reviewer for recognizing the analogy between the modeling of sequential editing and how humans repetitively revise their writings in real life. Other than that, we have also identified many applications that would benefit from the modeling of sequential editing. For sequential data such as sentences, Malmi et al. (2019) and Dong et al. (2019) have demonstrated that training models to produce a sequence of edits (rather than the edited sentence) leads to better performance on tasks such as sentence simplification and rephrasing. In addition, Welleck et al. (2019) and Gu et al. (2019a,b) have shown that modeling sequential editing enables text generation in arbitrary orders, so that a generative model can explore the most effective generation order on its own.
>
> In this paper, our focus is modeling sequential edits on _tree-structured data_. Similarly as sequential data, there are also applications where directly editing a tree-structured data is needed, e.g., editing the AST of computer programs given an edit specification (e.g., a code refactoring rule). Through experiments on two program editing datasets, we have shown that training a model to generate sequences of edits is advantageous than training it to generate the edited tree; it allows the model to learn more generalizable edit semantics than the alternative.
>
> **Novelty/contribution of our work**
>
> For a detailed discussion about the novelty/contribution of our work, please refer to **“To All Reviewers”**. In particular, while we define only four types of operator in this paper, the operators have been able to fulfill a broad range of editing requirements. Note that the operators are not fully allowed by other approaches (e.g., Brody et al. (2019) does not allow adding a new tree node).
>
> **Other questions**
> 1. “What is the advantage of doing these transformations prior to having the LSTM read the result?”: Since each type of operator involves different numbers of variables to characterize its information, we perform a transformation to project each of them into the same dimension so that they can be fed into one LSTM encoder.
> 2. We’re sorry for the oversight in not defining the “emb()” function -- to clarify:  all embeddings are learnable in our model.

---

### Author Response · Authors · 2020-11-15
**To All Reviewers: A summary of our contributions and the novelty of our proposed model**

We would like to thank all reviewers for their insightful comments!  We have attempted to answer questions below, and would be very happy to do any follow-up discussion or address any additional comments.

One of the concerns from several reviewers was regarding the novelty of the work. First, we apologize that it wasn’t crystal-clear in the submitted draft -- the paper had a lot of content and we had to remove some descriptions to meet the page limit. We will clarify it here and add the clarifications in the final draft later.

First, **the major contributions of our work** are as follows:
1. We study the problem of _iteratively_ editing tree-structured data, particularly triggered by an _edit specification_ (e.g., a code refactoring rule denoted by an edit pair <C-, C+>), which is an important task but has rarely been studied (except Brody et al. (2020), although the edit specifications we consider are still slightly different).
2. We propose a neural editor that runs iteratively to generate tree edits to a partially edited tree, which converts the editing problem into performing a sequence of incremental tree transformations. Our editor is language-agnostic and enforces grammar validity.
3. We propose a novel edit encoder to represent the edit specification by explicitly encoding the actual tree-level edits that are needed to convert C- to C+.
4. We propose an imitation learning algorithm for training our neural editor to correct its mistakes dynamically. As far as we know, this is the first attempt at imitation learning for non-sequential generative models.
5. Combining the technical contributions above, we have improved over previous state-of-the-art models on two standard program editing datasets, and have shown the potential of training editors to be robust to their likely mistakes.

Compared with existing approaches on the same or similar tasks, our study has novelty as discussed below:

**Comparison with Hoppity (Dinella et al., 2020)**: First, Dinella et al. are concerned with a different task setting than our contribution “1.”. Their study does not consider using an edit specification, and thus only performs _unconditioned_ tree edits, which is a major contrast to our work. As such, their model has no concept of our contribution “3.” (the edit encoder) and does not perform imitation learning as described in our contribution “4.”.

Second, in terms of the editor itself (our contribution 2.), we agree with reviewers that the idea of incremental tree transformations is similar to Hoppity’s in spirit. However, **our editor provides a more general framework owing to the adoption of ASDL.** By “general” we refer to the following two properties:
_(1) Language agnosticism._ While Hoppity also aims to be language-independent, it indeed has included JavaScript-specific designs. For example, its "REP_TYPE" operator for replacing the type of a non-terminal node is impossible for languages such as C#, for which revising a non-terminal node’s type (as well as its associated properties/fields) is not feasible.
_(2) Grammar validity._ ASDL specifies the number and types of child nodes that are syntactically valid for each tree node, which offers our editor holistic syntax guidance throughout the tree editing process (e.g., see ASDL’s mechanism for grammar validity after taking a Delete operator in Sec 3.1.1). In comparison, Hoppity’s grammar checking is ad-hoc; it is invoked only when needed at run time. Specifically, Hoppity applies grammar checking only when it selects the type of a non-terminal node.
We would like to emphasize that **incorporating ASDL (or any other grammar formalisms) into a tree editor is a non-trivial technical contribution** in terms of engineering effort and model designs (e.g. it required us to introduce the “dummy node” mechanism, as elaborated in Sec 3.1.1).

Finally, we would like to clarify our contribution “5.,” our empirical results. We first note again that Hoppity could not be applied to our setting due to the lack of an idea of conditioning on an edit specification, and JavaScript-specific grammar design. We would like to clarify our intention in mentioning that we are handling **“longer edit sequences”**: this was simply to stress that we evaluate our editor on a more challenging dataset (average number of edits: 7.264 on GHE and 7.089 on Fixers, when CopySubTree is allowed), while Hoppity was tested on datasets with up to 3 edits. We agree with Reviewer1 that the fact that Hoppity was evaluated with short edit sequences does not mean it cannot be applied to longer edit sequences, although we do hypothesize that our model design enforcing grammatical correctness may result in superior performance on longer edits. We will revise our argument regarding this.

... (see our follow-up comment)

---

> ### Author Response · Authors · 2020-11-15
> **To All Reviewers: A summary of our contributions and the novelty of our proposed model (Cont'd)**
>
> ... (continued comparison with Hoppity)
>
> While Hoppity is not directly applicable to our setting, our model can apply to Hoppity’s Javascript bug repair setting by (1) removing the edit representation, and (2) using an ASDL designed for JavaScript. **We will strive to add this experiment during the rebuttal period, but we will have to see whether it is feasible in the short time allowed.**
>
> **Comparison with Brody et al. (2020) and Tarlow et al. (2019)**: Again, we note that the problem setting that the two papers study is not the same as ours (our contribution “1.”): Tarlow et al. (2019) does not consider code edit pairs as edit specifications; while the setting of Brody et al. (2020) is close to ours, the edit specification they consider is the edits _in the context of the input tree_, rather than the desired edit pattern of _the input tree itself_. As a result, neither of them models the “edit encoder” (contribution “3.”) or can be directly applied to our datasets (contribution “5.”).
>
> Our editor’s model architecture is also very different from the two models’ (our contribution “2.”). Similarly to our editor, Brody et al. (2020) and Tarlow et al. (2019) produce edits rather than the edited trees. However, our editor’s architecture is distinct from theirs in the modeling of _sequential tree transformations_. Specifically, after generating each edit, we apply the edit to update/transform the tree and then re-encode the tree information for predicting the next edit. In contrast, Brody et al. (2020) and Tarlow et al. (2019) predict the complete sequence of edits based on the fixed initial tree, without applying its edits or performing any tree transformations. Owing to our tree transformation design, we further propose a novel imitation learning algorithm that works naturally to train our model to correct its mistakes, which is not provided by either of the two papers (our contribution “4.”).
> In addition, our editor supports a broader set of operators that are commonly used in tree editing. For example, we allow adding (or generating) new tree nodes while Brody et al. (2020) does not; compared with Tarlow et al. (2019), we additionally include a CopySubTree operator for efficient editing.
>
> Note: the work of Tarlow et al. (2019) is suggested by Reviewer 1, which as we understand refers to the paper -- “Learning to Fix Build Errors with Graph2Diff Neural Networks”, Tarlow et al., arXiv 1911.01205. We will cite this work in our revised draft.
>
> **We will follow up with individual responses shortly.**

---

### Author Response · Authors · 2020-11-24
**To All Reviewers: our revised paper has been uploaded**

We would like to thank all reviewers again for your thoughtful comments and discussions! We have updated our paper draft and have included clarifications and further analyses to address each reviewer's concerns. We will be very happy to address any additional comments.

---

### Decision · Program_Chairs · 2021-01-07
**Final Decision**

**Decision:**

Accept (Poster)

**Comment:**

This paper proposes a model for predicting edits to trees given an edit specification that comes either from the ground truth before-after state (“gold” setting, like reconstruction error of auto-encoder) or from the before-after state of an analogous edit. The problem setting follows mostly from Yin et al (2019).

There are several shortcomings of this paper:

1. The technical novelty of the model is somewhat limited, as it’s an assembly of components that have been used in related work. Authors insist in the discussion on the novelty of the tree edit encoder (Sec 3.2), but I think this is overstated. The related tree-edit models (e.g., Tarlow et al (2019)) perform a very similar encoding *in the decoder* when training with teacher-forcing. While it’s true that decoders are typically thought of as monolithic entities that generate a sequence of edits from a state, inside the teacher-forced training, the models are computing a representation of a prefix of ground truth edits, which are then repeatedly used to predict next edits. AFAIU, the proposal is basically to use this hidden representation as the edit encoder.

2. The claim that the approach is more language agnostic than Dinella et al (2020) also seems shaky, as the authors admit in their response that language-specific grammars need to be handled specially. E.g., I expect that the authors of Dinella et al would find it easier to extend their existing code to use a new language than to adapt this approach.

3. The submission relies too heavily on the “gold” setting (where the target output is fed as an input), and I’m skeptical of their characterization of Yin et al’s intentions when the authors say in comments, “Because of this, models that are able to reproduce the desired output effectively have a demonstrably better inductive bias that allows them to do so efficiently. This was the original motivation expressed by Yin et al. (2019).”  I don’t see this stated in the Yin et al paper. I see Yin et al. characterizing this setting as an upper bound and saying “better performance with the gold-standard edit does not necessarily imply better (more generalizable) edit representation.” (Yin et al., 2019). It’s worrying that the proposed model only seems to do better in this setting, which would be very easy to game if one were aiming to directly optimize for it.

Having said this, (1) is not a standard way to think about encoding edits, (2) is debatable, and we can hope that future work does not treat improvements in the “gold” setting as a valid research goal. Further, there is another contribution around imitation learning that the reviewers appreciate. In total, reviewers did an excellent job and generally believe the paper should be accepted. I won’t go against that recommendation.